# EMBRYOLOGY OF A LANGUAGE MODEL

## ABSTRACT

Understanding how language models develop their internal computational structure is a central problem in the science of deep learning. While susceptibilities, drawn from statistical physics, offer a promising analytical tool, their full potential for visualizing network organization remains untapped. In this work, we introduce an embryological approach, applying UMAP to the susceptibility matrix to visualize the model's structural development over training. Our visualizations reveal the emergence of a clear "body plan," charting the formation of known features like the induction circuit and discovering previously unknown structures, such as a "spacing fin" dedicated to counting space tokens. This work demonstrates that susceptibility analysis can move beyond validation to uncover novel mechanisms, providing a powerful, holistic lens for studying the developmental principles of complex neural networks.

## 1 INTRODUCTION

Just as developmental biologists seek to understand how a single fertilized cell gives rise to a complex organism with specialized organs, a central mystery in deep learning is how a randomly initialized network develops its intricate computational structures through training. In this work, we study the training process of language models as a form of *embryology*, where token sequences act as cells and their susceptibility vectors (Baker et al., 2025) serve as the analogue of gene expression profiles. By visualizing how a low-dimensional representation of these susceptibility vectors develops, we can watch as the model's "body plan" takes shape.

Given tokens $\Sigma$ and a language model parametrized by a neural network with components $C_1, \ldots, C_H$ (e.g. attention heads in a transformer) we associate a susceptibility vector

$$\eta_w(xy) = \left(\chi_{xy}^{C_1}, \ldots, \chi_{xy}^{C_H}\right) \in \mathbb{R}^H$$

to the combination of a weight $w$ for the network and a token sequence $xy$ where $x \in \Sigma^k$ is a context and $y$ is a possible next token. The entries in this vector are called *per-token susceptibilities* and they measure the covariance of two random variables, one depending on the component $C_j$ and the other on the continuation $y$ in context $x$. This vector of susceptibilities is sensitive to *how* the model computes its prediction of the next token given $x$. For example it was shown in Baker et al. (2025) that induction patterns (meaning token sequences like `the cat ... the `cat`` involving a bigram which is repeated in context, where the outlined token is $y$) tend to have a different pattern of susceptibilities across heads, and moreover this pattern can be used to distinguish the heads in the induction circuit (Olsson et al., 2022) across four seeds of a 3M parameter language model.

Given sequences $\{x_i y_i\}_{i=1}^n$ sampled from some distribution, for example the training distribution of our language model, the point cloud $\{\eta_w(x_i y_i)\}_{i=1}^n$ in $\mathbb{R}^H$ is a representation of the data distribution from the point of view of the model, since the configuration of token sequences under $\eta$ reflects patterns in how different components of the model interact to produce predictions of $y_i$ given $x_i$. Under the name *structural inference*, Baker et al. (2025) have proposed that studying such patterns is a means to discover and interpret the internal structure in neural networks.

As the next step in structural inference, it would be useful to visualize the image of the "data manifold" under $\eta$ in order to *see* this structure, and in particular, to see how it develops over training. Therefore in this paper, taking our cue from embryology where it is common to apply dimensional reduction techniques like UMAP (McInnes et al., 2020) to gene expression profiles in order to study the development of organisms (Cao et al., 2019), we propose to study the low-dimensional projection of

the point clouds $\{\eta_w(x_i y_i)\}_{i=1}^n$ under UMAP for weights $w = w(t)$ at various timesteps $t$ of training. The results are quite striking, as can be seen in Figure 1 (end of training) and Figure 2 (over training).

In this paper we make the following contributions. We

- **Introduce UMAP projections of susceptibilities** as an interpretability tool for studying the internal structure of language models and its development over training.
- **Show that the UMAP projection is stratified by token patterns**. The ontology of token patterns introduced in Baker et al. (2025) and recalled in Table 1 stratifies the UMAP, giving it the appearance of a "rainbow serpent" (Section 4.1).
- **Show that the emergence of the induction circuit** is visible in the serpent as a thickening of the dorsal-ventral axis as induction patterns (orange) separate from other patterns. This provides a new, and visual, perspective on a well-understood structural development in language models (Section 4.2) already studied using susceptibilities in Baker et al. (2025).
- **Show the emergence of a new structure** associated with the prediction of *spacing tokens* (e.g. actual spaces ⬚ but also newlines `\n` ) (Section 4.3).

Moreover, the same ontology of token patterns seems to be adopted across all four seeds of the language model in the sense that the same token patterns tend to occur in the same parts of the UMAP (Appendix F).

## 2 BACKGROUND

### 2.1 TOKENS AND PATTERNS

We denote by $\Sigma$ the set of tokens. For tokenization, we used a truncated variant of the GPT-2 tokenizer that reduced the original vocabulary of 50,000 tokens down to 5,000 (Eldan and Li, 2023). We denote token sequences as follows: ⬚wa ⬚vel ⬚ength is a sequence of three tokens. We often consider a token $y \in \Sigma$ in context $x \in \Sigma^k$. Following Baker et al. (2025) we consider eight *patterns* (Table 1) which are properties that either hold for individual tokens, or hold for tokens in a given context (possibly including subsequent tokens). Full definitions are given in Appendix A.

### 2.2 SUSCEPTIBILITIES

We define the susceptibility $\chi_{xy}^C$ for a component $C$ of a neural network used to predict the next token $y$ given a context $x$, and explain how to think intuitively about what these scalar values mean. For full details see Baker et al. (2025). We consider sequence models $p(y|x, w)$ that predict tokens $y \in \Sigma$ given sequences of tokens $x \in \Sigma^k$ for various $1 \le k \le K$ (called *contexts*) where $K$ is the maximum context length and $\Sigma$ is the set of tokens. The true distribution of token sequences $(x, y)$ is denoted $q(x, y)$. The sequence models we have in mind are transformer neural networks, where $w \in W$ is the vector of weights. We set $X$ to be the disjoint union of $\Sigma^k$ over $1 \le k \le K$ and $Y = \Sigma$.

Given a dataset $D_n = \{(x_i, y_i)\}_{i=1}^n$, drawn i.i.d. from $q(x, y)$ we define

$$\ell_{(x,y)}(w) = -\log p(y|x, w), \quad L_n(w) = \tfrac{1}{n}\sum_{i=1}^n \ell_{(x,y)}(w).$$

The function $L_n(w)$ is the empirical negative log-likelihood and its average over the data distribution is denoted $L(w) = \mathbb{E}_{q(x,y)}[\ell_{(x,y)}(w)]$. By a *component* of the neural network we mean some subset of the weights $C$ associated with a product decomposition $W = U \times C$. Given a parameter $w^* = (u^*, v^*)$ and writing $w = (u, v)$ for the decomposition of a general parameter, we define a generalized function on $W$ by

$$\phi_C(w) = \delta(u - u^*)\Big[L(w) - L(w^*)\Big] \tag{1}$$

where $\delta(u - u^*)$ is one if $u = u^*$ and zero otherwise. The *quenched posterior* at inverse temperature $\beta > 0$ and sample size $n$ is

$$p_n^\beta(w) = \frac{1}{Z_n^\beta}\exp\{-n\beta L(w)\}\varphi(w) \quad \text{where} \quad Z_n^\beta = \int \exp\{-n\beta L(w)\}\varphi(w)\,dw. \tag{2}$$

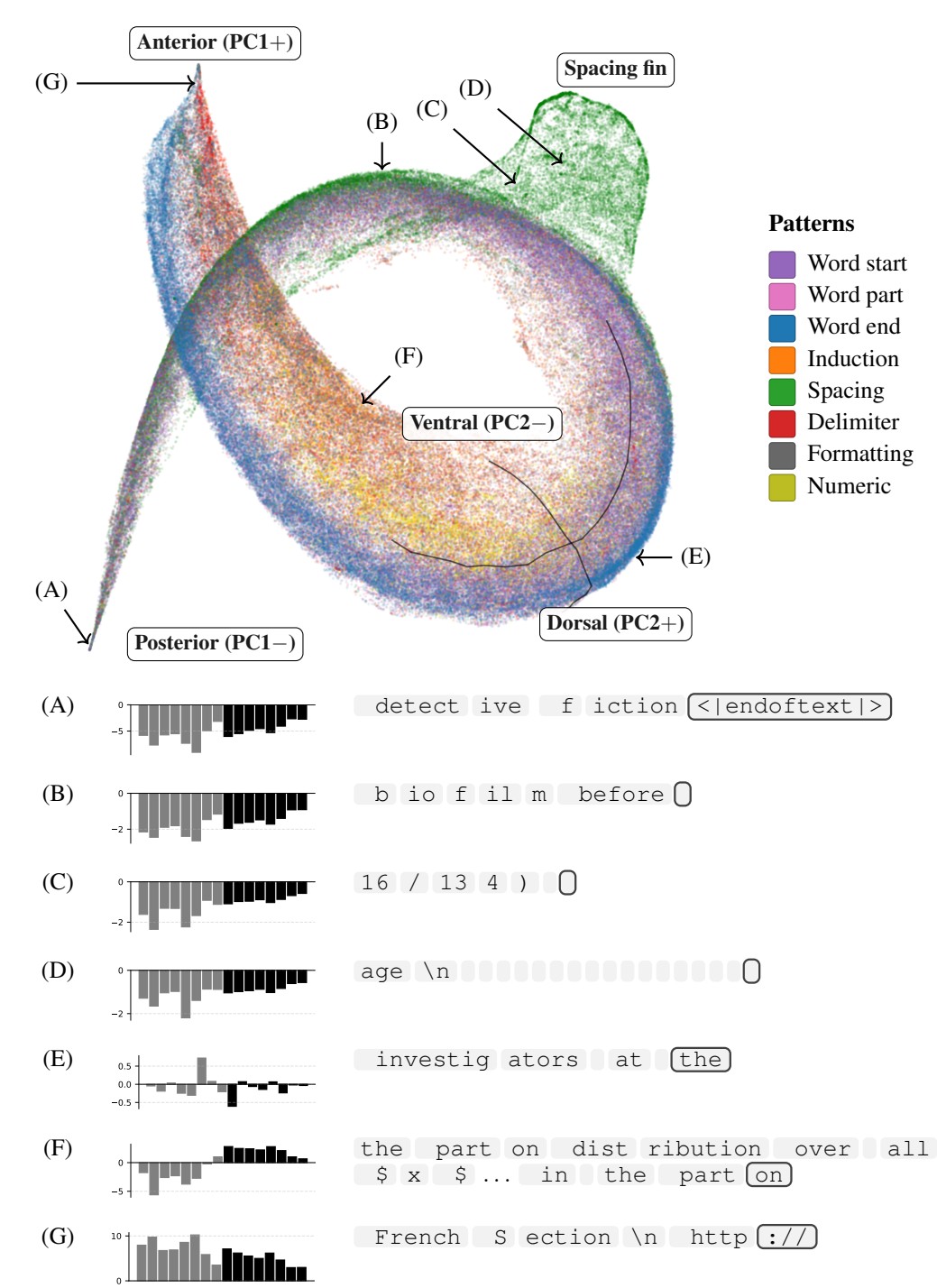

Figure 1: **The rainbow serpent: UMAP projection of susceptibility vectors.** (Top) Each point represents a token $y$ in context $x$, positioned according to its 16-dimensional susceptibility vector $\eta_w(xy)$ computed for a 3M parameter language model (one dimension per attention head) at the end of training (49900 steps). Token sequences $xy$ are colored by pattern, see Table 1. We mark the anterior-posterior (PC1) and dorsal-ventral (PC2) axes with black lines (Bottom) Susceptibility vector $\eta_w(xy)$ showing each head in order 0:0-0:7 (gray) 1:0-1:7 (black) and the token $y$ (black outline) in its context $x$.

| Pattern | Definition | Examples |
|---|---|---|
| **Word start** | A token that decodes to a space followed by lower or upper case letters | `be`, `R` ose , `The` |
| **Word part** | A non-word-end token that decodes to upper or lower case letters | S `ne` ed , `th` at , st `em` ed |
| **Word end** | A token that decodes to upper or lower case letters followed by a formatting token, delimiter or space | el im `inate` , differe `nces` ) , al `bum` |
| **Induction** | A sequence of tokens $uvUuv$ where $U$ is any sequence, $u, v$ are individual tokens, and $uv$ is not a common bigram ($q(v\|u) \leq 0.05$) | the cat ... the `cat` |
| **Spacing** | A token made up of one or more spaces, newlines, tabs, carriage returns, or form feeds | , \n , \t , \n\n |
| **Delimiter** | Brackets and composite tokens including parentheses, brackets, and their combinations | ) , ) , ] , ); , ( |
| **Formatting** | Tokens used for document structure and formatting beyond simple spacing | . , , , // |
| **Numeric** | Tokens containing numerical digits | 123 , 14 , 2024 |

Table 1: Token pattern categories and their definitions. Throughout the text we apply the indicated colors to tokens that follow a particular pattern.

Given a generalized function $\phi(w)$ we define the expectation

$$\langle \phi \rangle_\beta = \int \phi(w) p_n^\beta(w|h) dw. \tag{3}$$

and given a function $\psi(w)$ the covariance with respect to the quenched posterior is

$$\text{Cov}_\beta [\phi, \psi] = \langle \phi\,\psi \rangle_\beta - \langle \phi \rangle_\beta \langle \psi \rangle_\beta.$$

**Definition 2.1.** The *per-token susceptibility* of $C$ for $(x, y) \in X \times Y$ is

$$\chi_{xy}^C := -\text{Cov}_\beta \Big[\phi_C, \ell_{(x,y)}(w) - L(w)\Big]. \tag{4}$$

The susceptibility measures how $\phi_C(w)$ and $\ell_{(x,y)}(w) - L(w)$ covary when we perturb $w$ away from $w^*$, with perturbations being more likely according to their probability in the quenched posterior (that is, perturbations which increase the population loss $L(w)$ are exponentially suppressed). A variation in $w$ which only changes $L(w)$ by a small amount may nonetheless increase $\ell_{(x,y)}(w)$ for some tokens and lower it for others (e.g. a variation which improves performance on tokens in code, but worsens performance on poetry, might net out to a small overall change).

**Negative susceptibility** means that variations $w^* \to w$ which increase $\ell_{(x,y)}$ (that is, make $y$ less probable in context $x$) tend to be perturbations in the weights of $C$ which hurt the loss overall. This makes sense if $(x, y)$ follows a pattern that $C$ is involved in predicting, mechanistically. Thus we associate negative susceptibility with *the component $C$ expressing that $y$ should follow $x$.*

**Positive susceptibility** means that variations $w^* \to w$ which lower $\ell_{(x,y)}$ (that is, make $y$ more probable in context $x$) tend to be perturbations in the weights of $C$ which hurt the loss overall. This makes sense if $(x, y)$ follows a pattern that $C$ is involved in "opposing", mechanistically. It could be predicting an alternative completion, or just decreasing the probability of this one. Thus we associate positive susceptibility with *the component $C$ suppressing the continuation of $x$ by $y$.*

| Sign of $\chi$ | | Interpretation |
|---|---|---|
| $\chi_{xy} < 0$ | Expression | Variations in $C$ which decrease loss, also raise $p(y|x, w)$. |
| $\chi_{xy} > 0$ | Suppression | Variations in $C$ which decrease loss, also lower $p(y|x, w)$. |

For details on estimating susceptibilities see Appendix B.2 and Baker et al. (2025). In this paper, we use the same hyperparameters as given there.

## 2.3 STRUCTURAL INFERENCE

In both statistical physics (Altland and Simons, 2010) and biology there is a working definition of *structure* in a complex system that goes as follows: we take a set of *stimuli* (which might mean exposing the material to external fields in physics, or exposing a population of cells to chemicals or other environmental perturbations in biology) and a set of *observables* (which could be susceptibilities in physics, or gene expression levels in biology). This leads to a *data matrix* when we measure each observable after each stimulus. Analysis of this data matrix yields *modes of co-variation* of the material or cell in response to a common stimuli and these reveal functional structure in the system.

For example, in biology "gene co-expression networks based on the transcriptional response of cells to changing conditions" lead to the identification of *gene modules* which are "groups of genes whose expression profiles are highly correlated across the samples" (Zhang and Horvath, 2005).

The aim of *structural inference* as put forward in Baker et al. (2025) is to discover internal computational structure in neural networks from such analysis. The same "guilt-by-association" logic used in biology to identify gene modules applies to neural networks: identifying groups of components that respond in a coordinated fashion to data perturbations is a principled method for discovering their collective computational function.

## 3 METHODOLOGY

Our language model is the same 3M parameter attention-only transformer trained in Hoogland et al. (2025) and further studied in Wang et al. (2024); Baker et al. (2025). Unless specified otherwise, all results in the main text are for the same seed of the studied language model. Three additional models were trained using the same architecture and training distribution, but with different initializations and ordering of minibatches; see Appendix F other seeds.

This transformer has two layers, and in each layer only self-attention (no MLPs). Hence we refer to this as an *attention-only* transformer. The attention heads are denoted $l{:}h$ for $0 \leq l \leq 1$ and $0 \leq h \leq 7$. It is known from Hoogland et al. (2025) that the previous-token heads in this model are $0{:}1$, $0{:}4$, the current-token head is $0{:}5$ and the induction heads are $1{:}6$, $1{:}7$. This model was trained for 50000 steps on a subset of the Pile (Xie et al., 2023).

For a complete specification of architecture, including dimensions and training hyperparameters, please refer to Hoogland et al. (2025). While the model also contains embedding, unembedding, MLP and layer norm weights and it is possible to perform the same susceptibility analysis for these, we focus on attention heads in this paper.

## 3.1 UMAP

To each pair $(x, y)$ of a sequence of tokens $x$ and a token $y$ we associate a vector $\eta_w(xy) = (\chi_{xy}^1, \ldots, \chi_{xy}^H) \in \mathbb{R}^{\mathcal{H}}$ where $\mathcal{H}$ is a set of network components (e.g. attention heads) and $w$ is the parameter of the neural network. This can be thought of as a feature map which represents the continuation $y$ in context $x$ from the point of view of the model: token sequences will be mapped to nearby points in the feature space if the pattern of susceptibilities (e.g. the pattern of expression and suppression) across heads is similar. When we apply $\eta_w$ to transform a set of samples $(x, y) \sim q(x, y)$ from a given distribution over sequences and apply UMAP, this provides a low-dimensional visualization of this transformation of the data manifold. Note that the transformer does not receive $y$ as an input. See Appendix C for more on UMAP.

## 3.2 PER-PATTERN SUSCEPTIBILITY

Rather than focusing on individual tokens, we may aggregate the per-token susceptibilities by pattern. Let $\alpha$ be one of the pattern categories in Table 1. Given a set of contexts and a model parameter $w$ we can define the empirical *per-pattern susceptibility* $\hat{\phi}_w(\alpha) = \frac{1}{|\alpha|} \sum_{(x,y) \in \alpha} (\chi_{xy}^1, \ldots, \chi_{xy}^H)$ where

the sum is over those token sequences $xy$ in the sampled contexts that are classified as following the given pattern $\alpha$. We plot these values over training in Figure 2 to corroborate the qualitative observations in our UMAP visualizations.

# 4 RESULTS

We begin with a high level overview of the UMAP visualization and per-pattern susceptibilities (Section 4.1) before focusing more closely on two structures: induction (Section 4.2) and spacing (Section 4.3). The former relates to the development of the induction circuit of this model, which was previously studied in Hoogland et al. (2025); Wang et al. (2024); Baker et al. (2025) and offers a through-line for comparison of methodologies. The latter pattern indicates novel computational structure related to counting spacing tokens.

We show the development of the language model at four checkpoints: end of stage LM1 (step 900), end of stage LM3 (step 9000), end of stage LM4 (step 17500) and at the end of stage LM5 which is the end of training (step 49900). Notably, the induction circuit forms during stages LM3 and LM4. See Appendix B.1 for more details on these stages. In Appendix F, we present parallel experimental results for other seeds that demonstrate a high degree of universality in our observations.

In describing the axes of the UMAP body (see Figure 1), we use the *posterior-anterior* ("tail" and "head") and *dorsal-ventral* ("back" and "front") language from biology: the PC1 axis points from the posterior towards the anterior, and the PC2 axis from ventral to dorsal.

## 4.1 ONTOLOGY OF PATTERNS AND THE RAINBOW SERPENT

At the end of training the UMAP of susceptibility vectors in Figure 1 has a striking and colorful appearance when we color the points (i.e. token sequences) by pattern (Table 1). We refer to this as the *rainbow serpent*. In this section we explain why the UMAP appears this way, how it relates to the underlying structure of the model, and how the appearance develops over training.

Firstly, the rainbow serpent is a *serpent*: it is long and thin. This is because PC1 explains most of the variance. This is most pronounced near the beginning of training: PC1 explains 98.81% of the variance at step 900 and 95.19% at the end (Appendix C.2).

In Figure 1 we overlay black lines corresponding to the principal component directions, computed by constructing points along the lines spanned by each principal component and recomputing UMAP on these extrapolated points after computing it on the original data. We see that PC1 runs along the anterior-posterior axis, with positive PC1 pointing towards the anterior (all heads positive) while PC2 runs along the dorsal-ventral axis, with positive PC2 pointing towards what we have termed the dorsal side. As observed in Baker et al. (2025), tokens are distributed along PC1 according to their average susceptibility across heads: tokens that are broadly suppressed are at the anterior end, tokens that are broadly expressed are at the posterior. This is visible in the samples in Figure 1.

Secondly, we see that at the end of training the UMAP body is *stratified by color*: we note a clear blue stripe (word end tokens) along the dorsal side, a yellow streak near the mid-body running along the anterior-posterior axis (numeric tokens), purple towards the anterior (word starts), an orange "belly" (induction tokens from the mid-body to the ventral side), red near the posterior end (delimiters) and a body of spacing tokens extruding from the lower back (which we call the *spacing fin*). This is a refinement of the information we can derive from looking at large coefficient tokens in the principal components, as done in Baker et al. (2025) and reproduced here in Figure 5.

This stratification results from patterns in the tokens inducing differentiated patterns of susceptibilities across the heads; one natural explanation for this would be that potentially overlapping, but distinct, sets of heads are involved in expressing (or suppressing) each pattern. In this way the functional specialization of components of the model determines the "anatomical"* organization of the UMAP.

The organization of the serpent by color develops over training, as can be seen visually in the UMAP of Figure 2 and numerically in the accompanying plots of per-pattern susceptibilities over time. Up until the end of stage LM1 (900 steps) there is little stratification by token pattern. Rapid changes in

---

*Anatomy comes from the Greek ἀνατομή anatomē "dissection" or "to cut up" and as a technical term it seems quite apropos to the kind of analysis being performed here.

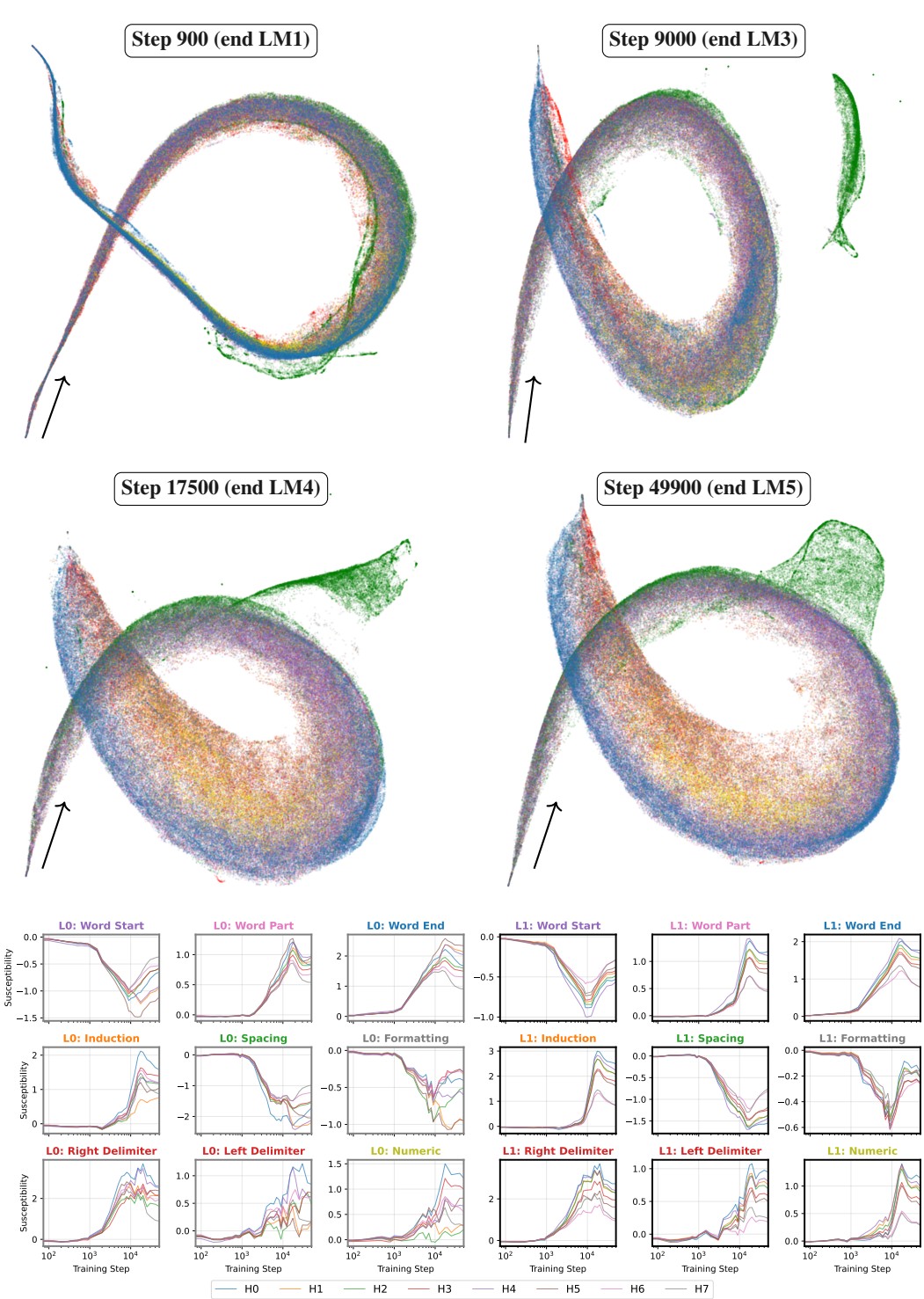

Figure 2: **Embryology of the rainbow serpent:** (Top) UMAP projections of per-token susceptibilities for all heads across training. Colors represent different token patterns, see Table 1. Arrows point from the posterior to the anterior. We select checkpoints according to the developmental stages, see Appendix B.1. (Bottom) Per-token susceptibilities for seed 1 are aggregated by pattern and their values are averaged and plotted over the course of training, using approx. 2M tokens. On the left half (gray outline) are the per-head, per-pattern susceptibilities for layer 0, while layer 1 susceptibilities are on the right (black outline). Training step ticks are shared across all subplots.

the per-pattern susceptibility occur over stages LM2 and LM3 (from 900 to 9000 steps). For example, we can see from the per-pattern susceptibilities that word start tokens have migrated to the posterior, and word end tokens have migrated to the anterior. Some of the spacing tokens are ejected from the main body during these stages (see Section 4.3 for more on the development of the spacing fin). Overall the main body remains "scrambled" in appearance at the end of stage LM3. During stage LM4 we see the emergence of the color stratification that characterizes the final model.

## 4.2 THICKENING AND INDUCTION

The importance of induction patterns and their role in the development of language models was first identified by Elhage et al. (2021); Olsson et al. (2022). The internal structure or *circuit* associated with the prediction of induction patterns is the *induction circuit*. An idealized induction circuit consists of a previous-token head in layer $l$ and an induction head in layer $l' > l$ that K-composes with the previous-token head; we refer the reader to Elhage et al. (2021) for the definition of K-composition. Induction heads are identified with the *prefix score* of Olsson et al. (2022). In practice, there can be multiple previous-token heads and induction heads cooperating to predict induction patterns. In the particular model we study, Hoogland et al. (2025); Wang et al. (2024) showed that the previous-token heads are 0:1, 0:4, the current-token head is 0:5 and the induction heads are 1:6, 1:7. We recall in Appendix I how Baker et al. (2025) identified the induction circuit consisting of these five heads using susceptibilities.

In the present paper, we use a data matrix which combines the heads from both layers. The loadings in Figure 15 are broadly positive in layer 0, negative in layer 1 and have the highest magnitude in layer 0 on the previous and current token attention heads, and the lowest (i.e., most positive) magnitude in layer 1 on the induction heads. This pattern also holds in the other three seeds (Appendix F.2). In this sense, the *induction circuit has a significant influence on the direction in susceptibility space* that points from the ventral to the dorsal side of the rainbow serpent.

The stratification of color along the dorsal-ventral direction in the UMAP, which runs from orange induction patterns on the "belly" to blue word endings on the "back", is made quantitative by Figure 5 and given the above remarks we can see that this stratification is a direct consequence of the fact that the *induction circuit expresses induction patterns* and the rest of the network tends to suppress them. In this way, an internal structure in the model, the induction circuit, appears both as a principal component of the matrix of susceptibilities and, relatedly, in the organization of token patterns in the anatomy of the serpent.

Moreover, the emergence of the induction circuit is represented visually in the development of the UMAP by thickening along the dorsal-ventral axis and the separation of induction patterns from other tokens (Appendix D).

## 4.3 EMERGENCE OF THE SPACING FIN

Near the beginning of training the spacing tokens (green) appear evenly distributed along the anterior-posterior axis. Around 1000 steps we see the per-pattern susceptibility for spacing tokens begin to decrease in both layers, signaling that these tokens are migrating to the posterior end, as can be seen in the UMAP from step 9000 onwards (Figure 2). Notably, at the end of LM3 we see a cluster of spacing tokens has separated from the main body (moreover, it remains separated even when we increase the `n_neighbors` hyperparameter of UMAP from 45 to 125, see Appendix C.1). This cluster, once it is reattached to the main body, will form the spacing fin that we see at the end of training. In Appendix E.1 we explain what distinguishes those spacing tokens that end up in the spacing fin.

This suggests that the model develops computational structure, possibly spread across many attention heads, for differentiating spacing tokens from one another and counting them. At present we lack a more mechanistic explanation of how this structure operates (analogous to the idealized model of the induction circuit).

## 5 RELATED WORK

**Representational geometry of language models.** A line of inquiry in interpretability has analyzed the geometry of language model representations. A landmark example is the work of Hewitt and Manning (2019) who introduced a "structural probe" to show that syntactic parse trees are geometrically embedded within BERT's activation space. Building on this, Reif et al. (2019) further explored this geometric organization. They applied UMAP to the context embeddings of polysemous words like "die," demonstrating that different word senses form distinct spatial clusters. Together, these works show how probing and visualization can map a model's representational manifold to known syntactic and semantic concepts. Important work on this topic has also been done in image models (Carter et al., 2019) and with SAE features (Bricken et al., 2023; Engels et al., 2025; Li et al., 2025). Our work adopts a similar visualization approach but shifts the focus: by applying UMAP to susceptibility vectors rather than activation vectors, we visualize information about the loss landscape geometry and how it relates to prediction. The geometry of representations has long been important in cognitive science (Kriegeskorte and Kievit, 2013).

**Expression and suppression.** The results presented in this paper further reinforce the picture in Baker et al. (2025) which suggests that the duality between expression and suppression may be fundamental to how computation in neural networks is organized. This is not entirely surprising, as exhibition / exhibition is known to be fundamental in *biological* neural networks. One theoretical justification for this has been put forward, that a strong and balanced exhibition/inhibition is necessary to calculate in the presence of noise in inputs and outputs (Rubin et al., 2017).

## 6 DISCUSSION

**Universal body plan.** Despite using different specific heads for tasks such as induction, the four training seeds of the language model end up with a strikingly similar overall structure (Appendix F). This suggests that while the low-level implementation (which head does what) is contingent, the high-level functional organization as revealed by the configuration in susceptibility space is to some degree a universal consequence of the architecture and the data distribution. It is an interesting question for the science of deep learning whether the universal principles of organization behind such empirical observations can be clarified.

**Limitations.** Our study has several limitations, which present important directions for future work. The primary limitation is the scale of the model: a 3M parameter, two-layer attention-only transformer. Whether the UMAP of susceptibility vectors is useful for discovering structure in larger language models remains an open question. Second, our interpretability method relies on UMAP for visualization. Although we focused on robust, large-scale phenomena like the separation of token patterns, UMAP can distort global distances, and thus the geometry of the "rainbow serpent" should be interpreted cautiously. Finally, the structures we identified may be contingent on our experimental setup; for instance, the prominence of the spacing fin is likely influenced by the tokenizer, and different tokenization strategies could lead to different learned structures. It is therefore important to investigate these phenomena across different model scales and tokenizers.

## 7 CONCLUSION

In this work, we have demonstrated that the development of language models can be usefully studied through an embryological lens, by applying UMAP to visualize the evolution of susceptibility patterns across training. We observed the emergence of clear anatomical organization: an anterior-posterior axis defined by global expression versus suppression, dorsal-ventral stratification corresponding to the induction circuit, and the formation of a new structure that we call the spacing fin.

Moreover, the remarkable universality of the developmental trajectories across different model seeds suggests that language models may be discovering fundamental organizational principles dictated by their architecture and data distribution.

## 8 REPRODUCIBILITY STATEMENT

Appendix B outlines experimental details of the model and of hyperparameters used in sampling. Appendix C covers the UMAP hyperparameters used, and Appendix G lists hyperlinks to the datasets used. Further implementation details and theoretical treatment can be found in Baker et al. (2025).

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

APPENDIX

The appendix provides supplementary material to support and expand upon the main text. It is organized as follows:

- **Appendix A:** full details on token pattern definitions.
- **Appendix B:** experimental details, including developmental stages, and posterior sampling hyperparameters.
- **Appendix C:** UMAP hyperparameter details, discussion on the specific tradeoffs of PCA versus UMAP as dimensionality reduction techniques, and additional plots relating to the dorsal-ventral thickening of the UMAP serpent.
- **Appendix D:** additional information related to the induction circuit and its representation in the UMAP.
- **Appendix E:** additional plots relating to the spacing fin.
- **Appendix F:** developmental UMAP serpents and per-pattern susceptibilities for three other model seeds.
- **Appendix G:** descriptions and links to the datasets used.
- **Appendix H:** additional viewing angles for the UMAP serpent, revealing otherwise hidden token structures.
- **Appendix I:** additional related work.
- **Appendix J:** a statement on the use of LLMs in this research.

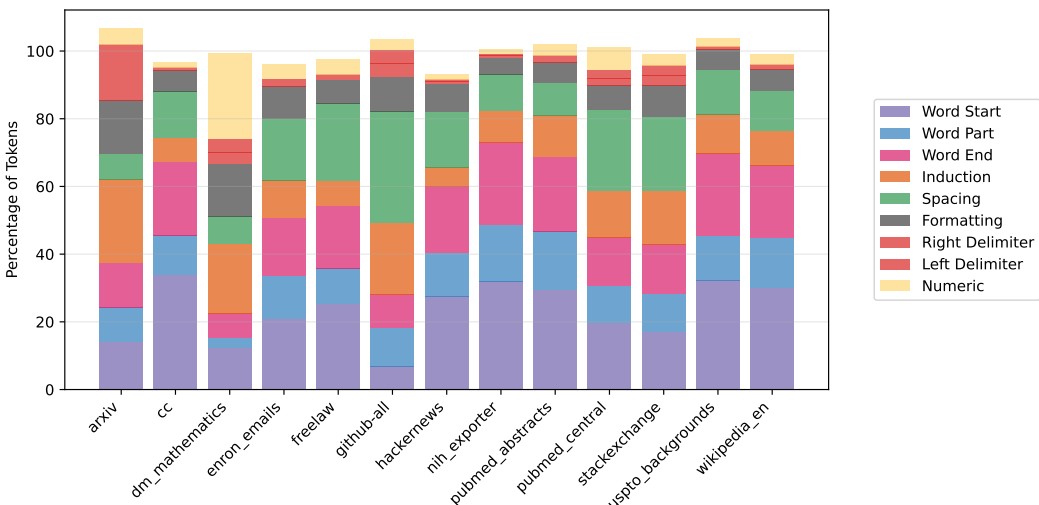

Figure 3: Percentages of tokens in each dataset which follow a given pattern. Note that not all patterns are mutually exclusive.

# A    TOKEN PATTERN DEFINITIONS

A summary of the token patterns used in this paper is given in Table 1. In the rest of this section we give a fully detailed definition of each pattern. Let $\Sigma$ denote the set of tokens in our tokenizer.

**Definition A.1.** A *left delimiter token* is an element of the set of tokens

$$\left\{ \text{<}, \text{ <}, \text{\{}, \text{ \{}, \text{(}, \text{ (}, \text{[}, \text{ [}, \text{</}, \text{\{"}, \text{ \$} \right\}.$$

A *right delimiter token* is an element of the set of tokens

$$\left\{ \text{>}, \text{ >}, \text{\}}, \text{ \}}, \text{)}, \text{ )}, \text{]}, \text{ ]}, \text{),}, \text{],}, \text{):}, \text{).}, \text{))}, \text{);}, \text{\%)}, \text{\$} \right\}.$$

We call a token a *delimiter token* if it is either a left or right delimiter token.

The asymmetry between left and right delimiters is due to the tokenizer. For our model, right delimiters seem much more important than left delimiters.

**Definition A.2.** A *formatting token* is an element of the set of tokens

$$\left\{ \text{~}, \text{\\}, \text{ \\}, \text{/}, \text{//}, \text{ //}, \text{://}, \text{-}, \text{ -}, \text{--}, \text{ --}, \text{\_}, \right.$$

$$\text{========}, \text{\_\_}, \text{----}, \text{--------}, \text{-----------------}, \text{**}, \text{****}, \text{*******},$$

$$\text{\#\#\#\#}, \text{.}, \text{,}, \text{:}, \text{::}, \text{ :}, \text{;}, \text{ ;}, \text{",}, \text{<|endoftext|>},$$

$$\left. \text{="}, \text{":"}, \text{|}, \text{'}, \text{"}, \text{->}, \text{ ->}, \text{\^}, \text{\%} \right\}.$$

**Definition A.3.** A *word start* is a single token that begins with a space and is followed by lower or upper case letters. That is, it is a token which when de-tokenized matches the regular expression `" [A-Za-z]+$"`.

**Definition A.4.** A *spacing token* is a token which when de-tokenized is a sequence of characters from the set

$$\left\{ \text{ }, \text{\n}, \text{\t}, \text{\r}, \text{\f} \right\}.$$

**Definition A.5.** A *numeric token* is a token which when de-tokenized and with spaces removed, consists of one or more digits.

The patterns defined above are independent of the context in which a token appears. By contrast, the subsequent definitions apply to a token in a given context.

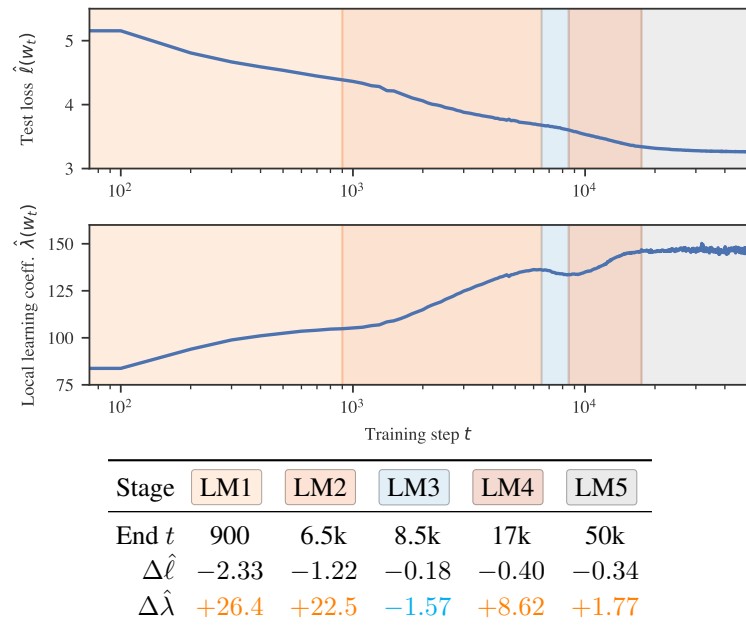

Figure 4: **Developmental stages of the language model.** Critical points in the local learning coefficient (LLC) curve mark boundaries between distinct *developmental stages* (bottom row; warm hues for increasing LLC, cold for decreasing LLC).

**Definition A.6.** A *word end token* is a token which when de-tokenized is made up of upper or lower case letters and which is followed in its context by a single formatting token, delimiter or space.

**Definition A.7.** A *word part token* is a token which is not a word ending in its context and which when de-tokenized consists of upper or lower case letters.

**Definition A.8.** An *induction pattern* is a sequence $xyUxy$ where $U \in \Sigma^*$ and $x, y \in \Sigma$, satisfying the following conditions:

- The conditional probability of $y$ following $x$ satisfies $q(y|x) \leq 0.05$.

- $x, y \notin \{$ ` `, `\n`, `,`, `.`, `the`, `to`, `:`, `and`, `by`, `in`, `a`, `be` $\}$.

Note that $U$ can be the empty sequence and may contain occurrences of $x, y$. In a given context we classify a token as an *induction pattern token* if it is $y$ for an induction pattern $xyUxy$ within the context.

We use estimated conditional probabilities based on samples from the Pile. Note that the sets of left delimiters, right delimiters, formatting tokens, word start tokens and word part tokens are pairwise disjoint. The set of induction pattern tokens and word part tokens are disjoint. The percentage of $N = 20000$ tokens sampled from each dataset which fit each of these patterns are given in Figure 3.

Note that the structure learned by a model may be substantially influenced by the tokenizer: for instance, the importance of space tokens in this paper is partly explained by the fact that many sentences are tokenized with individual spaces ` ` that are not bound into words, as for example the space in ` The ` is bound.

## B EXPERIMENTAL DETAILS

### B.1 DEVELOPMENTAL STAGES

In Figure 4 we recall the five developmental stages of the language model as discovered in Hoogland et al. (2025):

- **LM1** : The model learns to predict common bigrams, patterns of token pairs $x \rightarrow y$.

- **LM2** : The model learns to predict common multigrams, patterns of tokens $x_1 U_1 \cdots x_k U_k \rightarrow x_{k+1}$ where $U_i$ may be empty or not (i.e. they may be "skip" multigrams or not).

- **LM3** : The previous token heads begin to form.

- **LM4** : The induction heads begin to form.

- **LM5** : Stage five was identified as a distinct stage by Hoogland et al. (2025), but the primary developmental changes were not determined.

### B.2 SUSCEPTIBILITIES HYPERPARAMETERS

Following Baker et al. (2025), we compute susceptibilities using Stochastic Gradient Langevin Dynamics (SGLD) with hyperparameters $\gamma = 300$, $n\beta = 30$, $\varepsilon = 0.001$, batch size 16, 4 chains, and 100 draws to compute the per-token susceptibilities.

For additional details on the theory and implementation of susceptibilities used in this paper, please refer to the appendices of Baker et al. (2025).

## C UMAP

We chose UMAP for dimensionality reduction because of its speed and accurate representation of local topological properties of the dataset. Any method of dimensionality reduction introduces potential for error into an analysis, because there is simply no way to accurately reflect the geometry of a high dimensional space in two or three dimensions. Some common faults are that UMAP does not preserve global structure, and that it represents data as roughly uniformly dense even when that is not the case. See Chari and Pachter (2023) for a more detailed discussion.

To avoid this, we focus our analysis on observations where it is clear that UMAP has faithfully represented an aspect of the original distribution. The phenomena analyzed in this paper happen at large scale, and are ultimately determined by which points are near each other (in both the UMAP plot and high-dimensional susceptibility space), so we can be confident they are not illusions.

We apply UMAP to a data matrix $X$ with 16 columns and $260k$ rows. Each row is the susceptibility vector $\eta_w(xy)$ for a fixed neural network parameter $w$ where $(x, y) \sim q^l(x, y)$ as $1 \leq l \leq 13$ ranges over the datasets in Appendix G except for PILE1M. We sample $20k$ token sequences for each dataset. The data matrix $X$ is standardized (that is, the columns have the mean subtracted and are rescaled to have unit standard deviation) before applying the UMAP algorithm.

### C.1 UMAP HYPERPARAMETERS

The UMAP algorithm depends fundamentally on the choice of `n_neighbors` hyperparameter. The images in this paper were computed with `n_neighbors`$= 45$.

This hyperparameter governs how many nearest neighbors are taken into consideration when computing the local distances in the original embedding that the learned embedding tries to match. The value being too low can cause misleading clusters of data points in the visualization. Since our analysis includes identifying clusters, we took pains to identify such false patterns. We rendered visualizations with `n_neighbors` ranging between 15 and the equivalent of 8000 (via down-sampling by a factor of 8, with `n_neighbors` set to 1000), and any phenomena that were not observable in all renderings were dismissed as spurious.

We also created the plots with a range of values between 0 and 0.5 for the `min_dist` parameter, which has a maximal value of 1. This value governs how close neighboring points are allowed to be when learning the low dimensional embedding. We did not see substantial differences in the plots for different `min_dist` values. The figures in this paper were rendered with `min_dist` $= 0.1$.

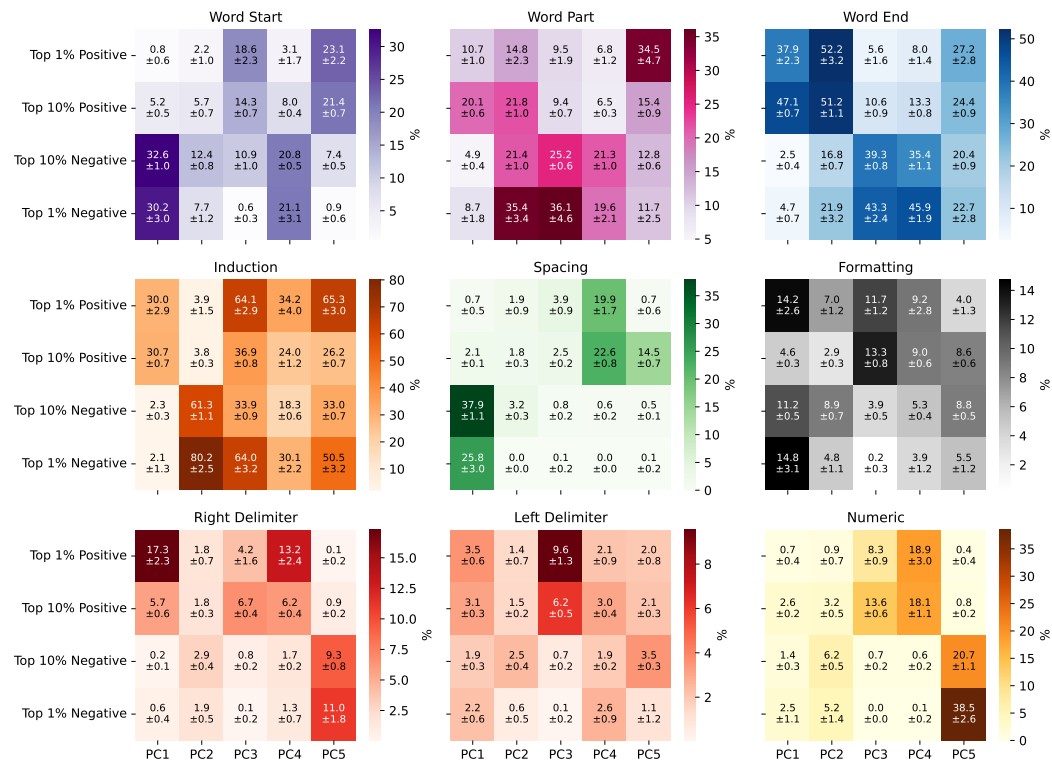

Figure 5: **Per-token susceptibility PCA** showing mean and standard deviation of loadings of principal components on data patterns across 10 independent draws of 20000 tokens from each dataset. The data matrix is the same as that used for UMAP, see Appendix C. The number 30.2 appearing in "Top 1% Negative" for word start tokens in PC1 means that when we order the tokens in PC1 with negative coefficients by magnitude, 30.2% of those in the top 1% by magnitude are word start tokens.

## C.2 UMAP VERSUS PCA

Throughout this paper, UMAP is our dimensionality reduction tool of choice. Any dimensionality reduction tool comes with tradeoffs, and we discuss our reasons for choosing UMAP in section Section 3.1.

However, the most common such tool is PCA, so it is natural to ask why we do not also incorporate PCA visualizations into our analysis. For PCs 1 and 2, we already capture much of the information from the fact that PC1 runs along the posterior-anterior direction and that PC2 runs along the dorsal-ventral direction. Together, these capture around 98% of the explained variance. Below, we present the explained variances for principal components 1 to 6:

- PC1: 0.9519 (0.9519 cumulative)

- PC2: 0.0262 (0.9781 cumulative)

- PC3: 0.0058 (0.9839 cumulative)

- PC4: 0.0033 (0.9872 cumulative)

- PC5: 0.0028 (0.9900 cumulative)

- PC6: 0.0022 (0.9922 cumulative)

We see that for PC3 and beyond, the amount of explained variance is comparable for the different principal components, and so it is not obvious that privileging PC3 above 4, 5, and 6 in a visualization gives as useful of a representation of the susceptibilities as a UMAP reduction does.

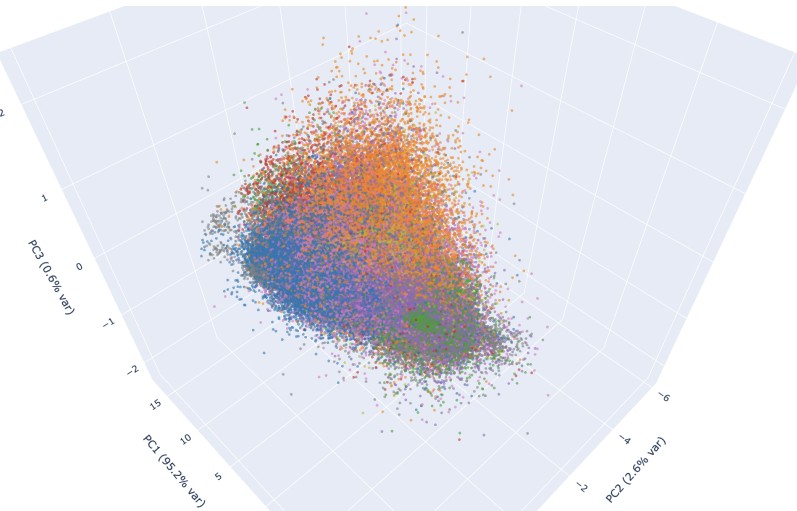

Figure 6: A 3D visualization of the first three principal components is shown for the same per-token susceptibility data used to generate the UMAP serpents. Note that the different axes were automatically rescaled.

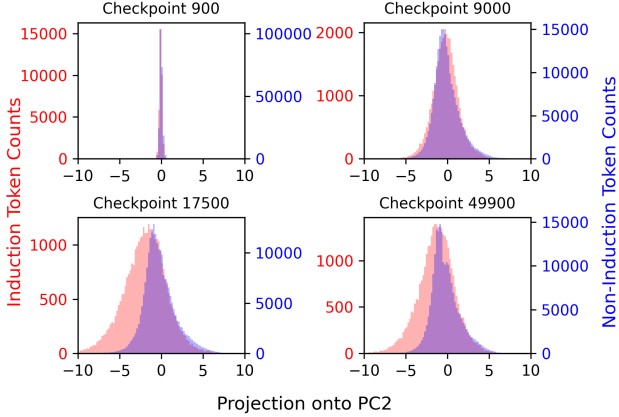

Figure 7: The distribution of tokens projected onto PC2 in susceptibility space, separated into induction pattern tokens and all other tokens.

As an example, in Figure 6, we have a 3D plot of the first three principal components of the susceptibilities data. Some of the macroscopic structure visible in the UMAP serpent, such as the general posterior-anterior organization and the separation in PC2 of induction pattern tokens. However, the spacing fin is no longer visible: the tokens composing the spacing fin in the UMAP plot now make up the densest visible cluster of green tokens in the center of the head. This suggests that the structural information that separates the spacing fin is contained in PC4 and higher, which is lost by using PCA as our dimensionality reduction technique but preserved by UMAP. Similarly the streak of numeric tokens visible in the UMAP (Appendix H) appears only in PC5 (see Figure 5). Therefore, for the type of analysis we are doing in this paper, the tradeoffs in using UMAP are preferable.

In Figure 5 we show the percentage of extreme positive and negative tokens in the PCs which follow a given pattern; this gives a quantitative picture of the extremes of the point cloud of susceptibility vectors.

### C.3 THICKENING OF PC2

In Figure 7, we see that between checkpoints 9000 and $17,500$, the induction pattern tokens grow significantly relative to other tokens along PC2 and maintains this through the end of training. As stated in the main text, the explained variance for PC2 increases from $2.3\%$ to $6.4\%$ between steps 9000 and $17,500$. This quantifies the visual striation and thickening of the induction tokens visible in Figure 1, showing that this is not just a misleading visual artifact of UMAP.

## D INDUCTION CIRCUIT

In this section we study how the development of the induction circuit is represented in the evolution of the structure of the UMAP. This development was originally studied in Hoogland et al. (2025), see Appendix I. In the present paper the most natural indicators of development are the per-pattern susceptibilities in Figure 2 for induction patterns, which in both layers are small and negative until around 2000 steps when they begin a slow increase, which becomes rapid sometime before 10000 steps. The effect of this increase is visible in the UMAP: at 9000 steps there is no dorsal-ventral stratification by color, whereas this is obvious at 17500 steps (the end of LM4). The thickening of the serpent is explained by the variance of the susceptibility of induction tokens along the direction of PC2 experiencing a sharp increase: the percentage of total variance of the subset of induction tokens accounted for by PC2 increases from $2.3\%$ to $6.4\%$ (see Appendix C.3 for more details) between 9000 and 17500 steps.

## E SPACING FIN

### E.1 DEVELOPMENT

In Figure 8, we show the development of the spacing fin over training. We see that these are token sequences $xy$ where $y$ is a spacing token and $x$ *ends with a sequence of spacing tokens* (typically a space preceded by a sequence of spaces, as in the examples (B)-(D) of Figure 1). By the end of training we can see that moving outwards along the spacing fin from the UMAP body we encounter (on average) token sequences where $y$ is preceded by an increasing number of spacing tokens; in particular, some tokens on the rim of the spacing fin exist in a "desert" of hundreds of spaces. This observation is made quantitative by Figure 11 which also indicates the approximate direction in susceptibility space that the spacing fin juts out towards.

Another interesting development phenomenon is the the reattachment of the spacing fin to the main body, first to the posterior end at 17500 and then to the anterior end sometime before the end of training. As can be seen in Figure 8 and the examples of Figure 1 the reattachment at the posterior end is easy to understand: the ordinary spaces (preceded by non-spacing tokens in context) that make up the posterior bulk of spacing tokens (see Figure 9) become "glued" to the token sequences $xy$ in the fin where $x$ ends in a small number of spaces. The anterior attachment of the spacing fin is more subtle, since the region it attaches to contains newlines `\n` rather than spaces.

### E.2 ADDITIONAL FIGURES

In Figure 9, we see the distribution of spaces and newlines in the serpent roughly split between the anterior and posterior halves, and we also see the spacing fin attached at the boundary.

In Figure 10, we see a histogram of the frequency of different counts of consecutive spacing tokens. Note the log scale on the y-axis, indicating roughly exponential decay.

In Figure 11, we see the evolution of per-head susceptibilities on spacing tokens as the minimum number of preceding spacing tokens increases, indicating directions in susceptibility space that increasing consecutive spacing tokens points in.

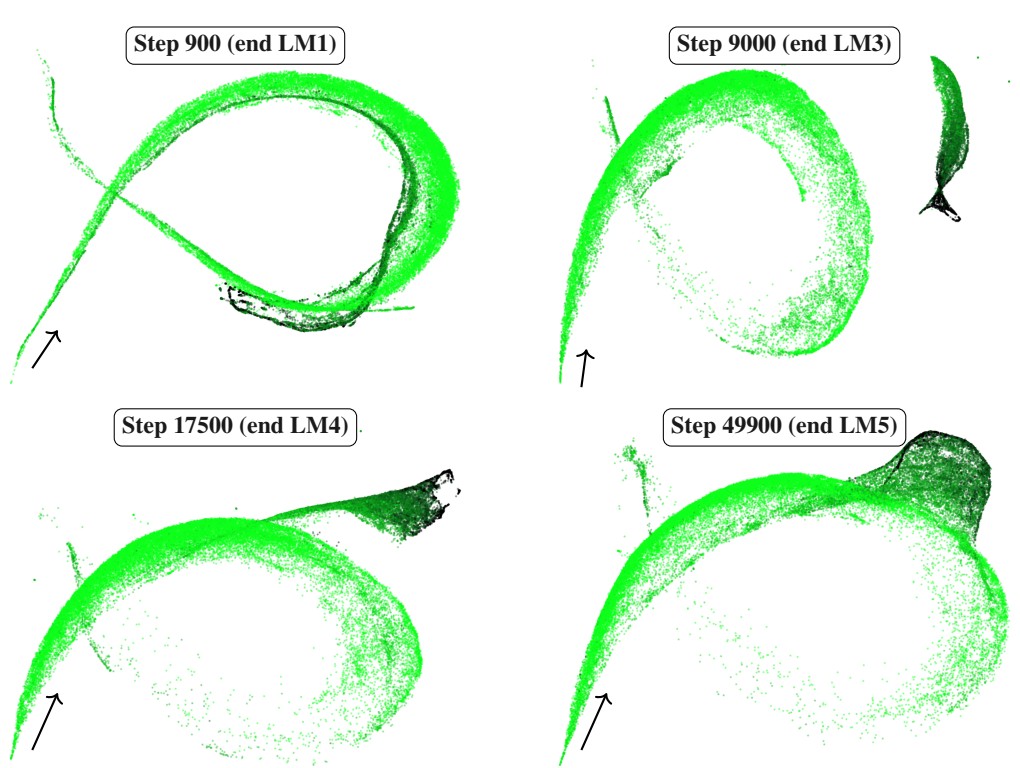

Figure 8: **The development of the spacing fin.** UMAP projections of spacing tokens across training using approximately 42000 tokens. Tokens are colored according to the number $s$ of preceding spacing tokens with green component $1 - 0.8a$ where $a = \log_{10}(1 + s)/\log_{10}(1 + 50)$. Points with 50 or more preceding spacing tokens are black, with the maximum number of preceding spacing tokens being 665. Arrows indicate the direction from the posterior to the anterior.

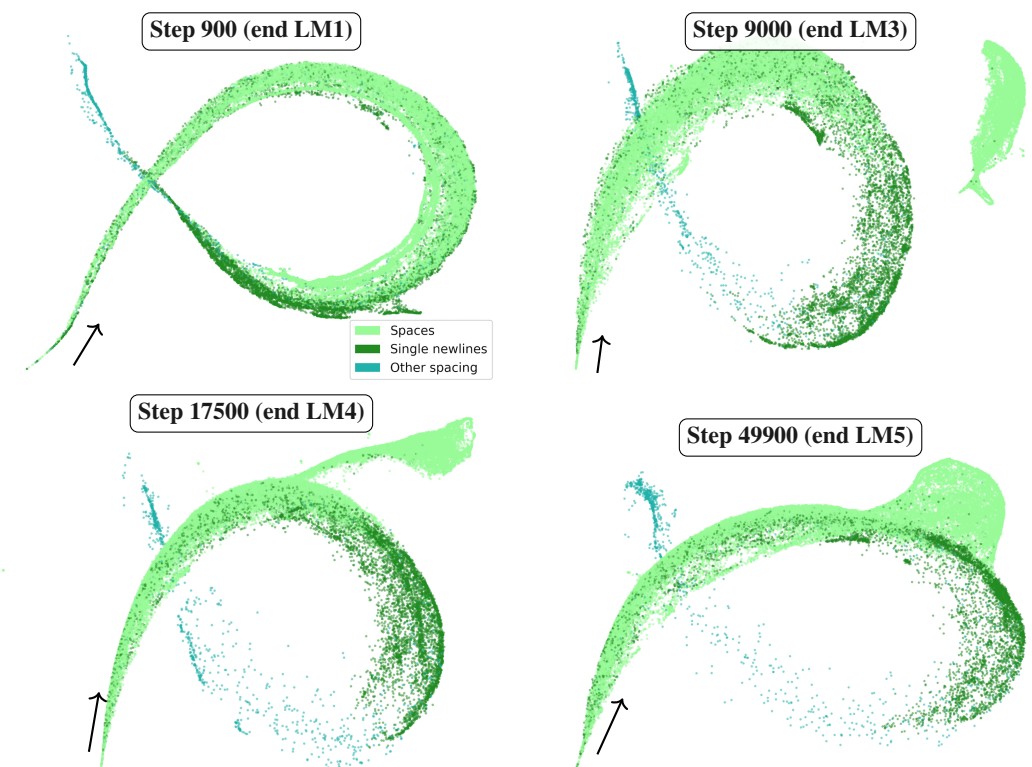

Figure 9: **The development of the spacing fin, by type.** UMAP projections of spacing tokens at four checkpoints across training, using approximately 42000 tokens. Tokens are colored according to whether they are single spaces ⬚ , individual newlines `\n` or other. Susceptibilities for both layers are used, and arrows indicate the direction from the posterior to the anterior.

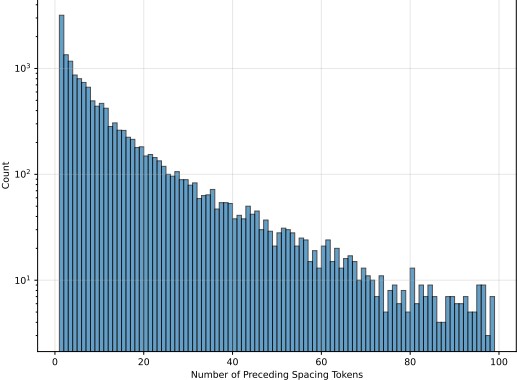

Figure 10: Histogram of the number of spacing tokens preceding a spacing token, in the approximately 42k spacing tokens found in the set of 260k sampled token sequences.

## F  OTHER SEEDS

Four models were trained in the same setting as described in Hoogland et al. (2025). In the main text we describe results for seed 1 and in this appendix we provide partial results for other seeds (2, 3, 4). Note that we use the same checkpoints for direct comparison to seed 1 but the stage boundaries for each seed differ by small amounts.

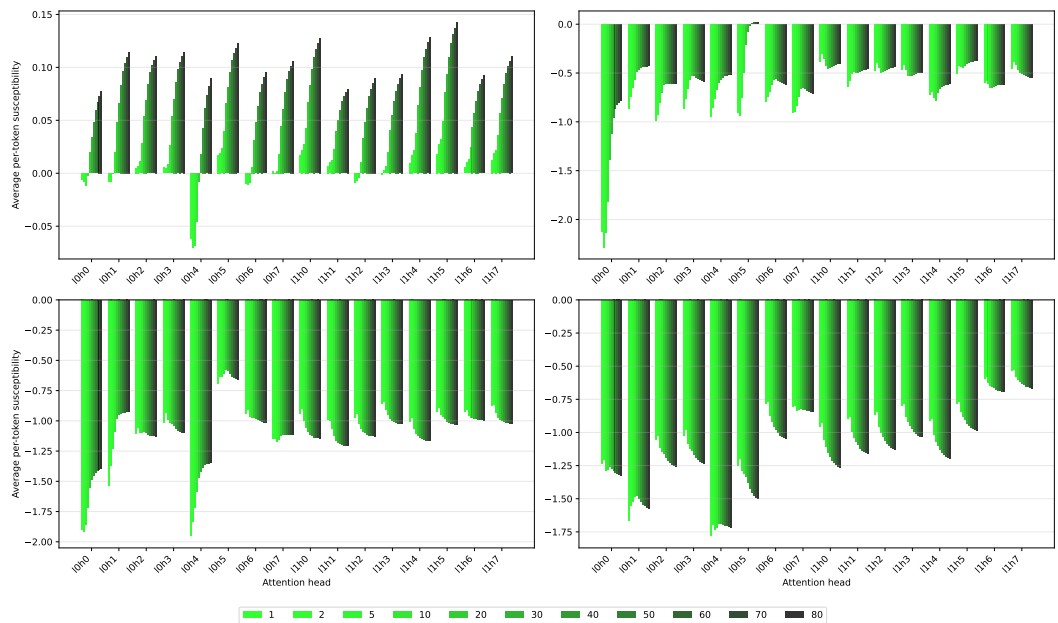

Figure 11: The average per-token susceptibilities for spacing tokens is shown across four checkpoints (900, 9000, 17500, 49900 reading from left to right and top to bottom) conditional on the minimum number of preceding spacing tokens. Colors range from green (1) to black (80).

## F.1 UMAPS AND PER-PATTERN SUSCEPTIBILITIES FOR OTHER SEEDS

The UMAPs are shown in Figure 12, Figure 13, and Figure 14. Across these different seeds, we see an astonishing degree of universality in the macroscopic structure of these models, including the ways particular patterns cluster towards the anterior or posterior, the dorsal-ventral striations, and the existence of a spacing fin. The most obvious difference appears to be the specific stages of development that the spacing fin undergoes, with seed 2 seeing a similar ejection of tokens from the main body, while seeds 3, 4 do not.

We did not investigate other seeds as closely as seed 1, so we do not know the source of these differences. We do note that there is some tension in desiring universality, while still wanting to be able to pick up on meaningful differences between seeds.

The similarities in UMAP visualizations are reflected in the accompanying per-pattern susceptibilities plots. Across the different seeds and patterns, the overall shape of the curves is fairly consistent, with the largest differences occurring after the curves begin to fan out and differentiate from each other. Even the differentiations share many qualitative features, for example, the shapes of induction pattern susceptibilities for layer 1 heads are highly regular.

## F.2 PC LOADINGS

In Figure 16-Figure 18 we show the loadings on heads of the top three principal components for the data matrix $X$ of susceptibilities.

## G DATASETS

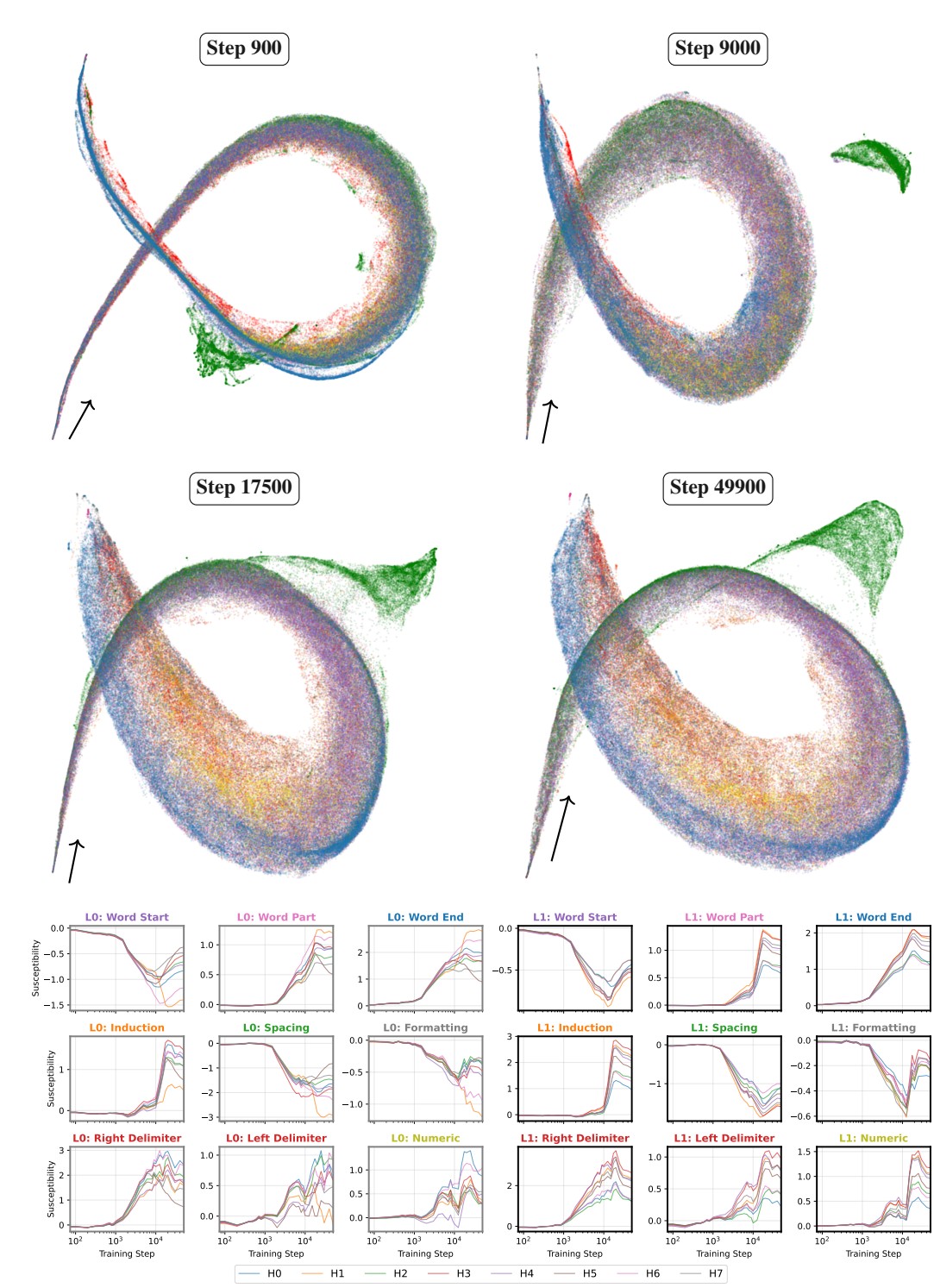

Figure 12: **Embryology of the rainbow serpent, seed** 2**:** (Top) UMAP projections of per-token susceptibilities for all heads across training for seed 2. Arrows point from the posterior to the anterior. (Bottom) Per-token susceptibilities for seed 2 are aggregated by pattern and their values are averaged and plotted over the course of training, using approximately 2M tokens. On the left half (gray outline) are the per-head, per-pattern susceptibilities for layer 0, while layer 1 susceptibilities are on the right (black outline).

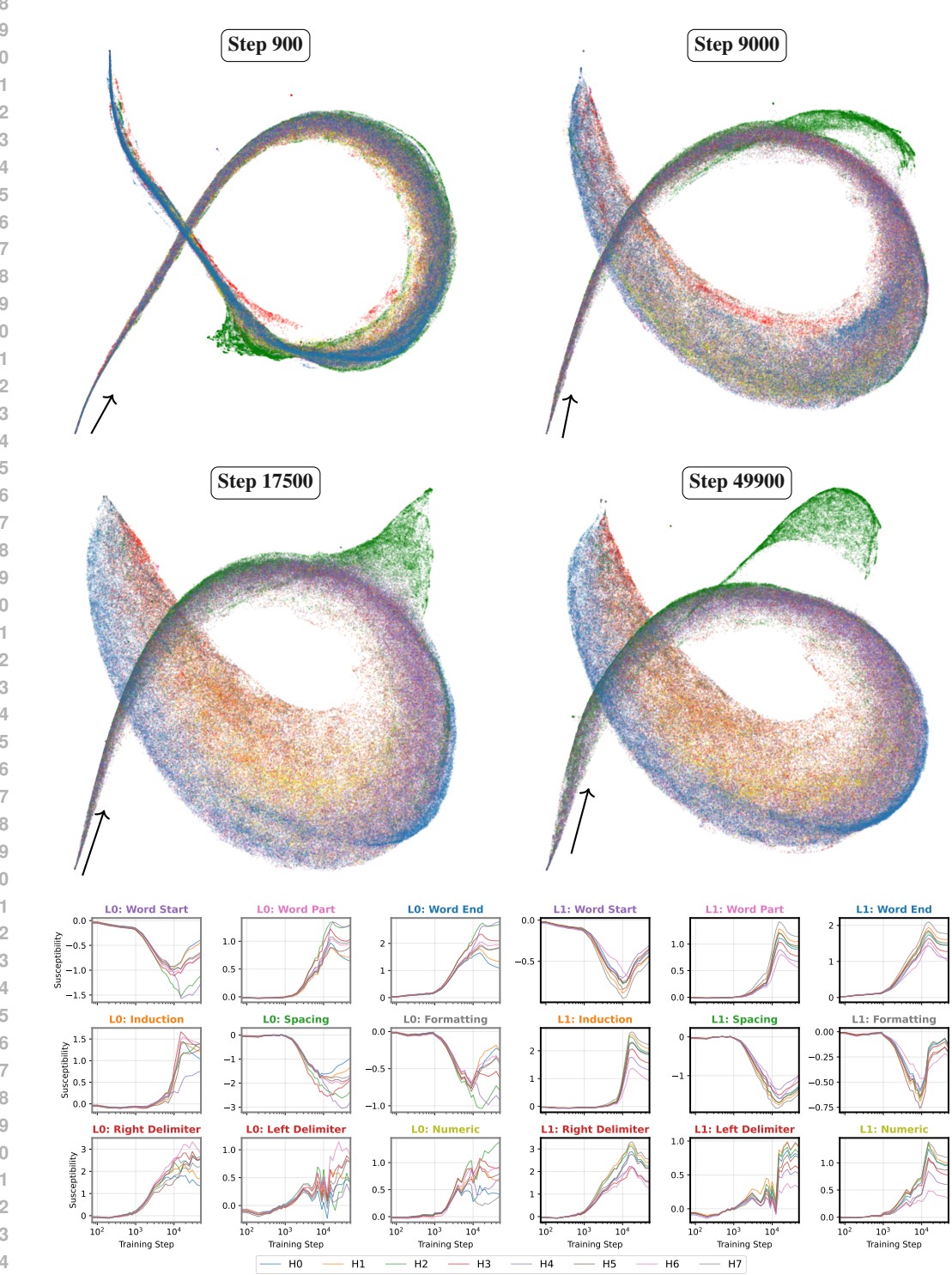

Figure 13: **Embryology of the rainbow serpent, seed** 3: (Top) UMAP projections of per-token susceptibilities for all heads across training for seed 3. Arrows point from the posterior to the anterior. (Bottom) Per-token susceptibilities for seed 3 are aggregated by pattern and their values are averaged and plotted over the course of training, using approximately 2M tokens. On the left half (gray outline) are the per-head, per-pattern susceptibilities for layer 0, while layer 1 susceptibilities are on the right (black outline).

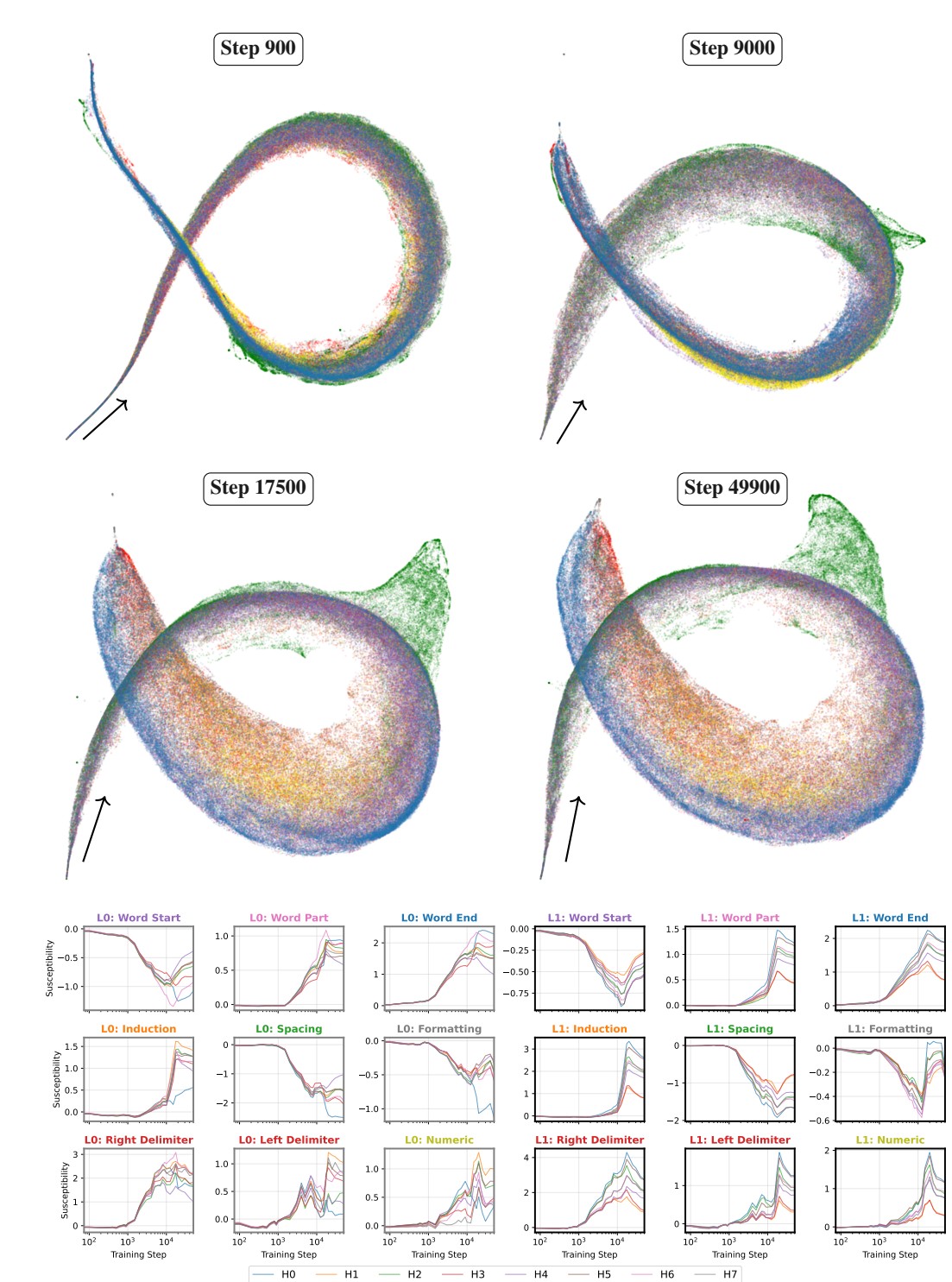

Figure 14: **Embryology of the rainbow serpent, seed** 4**:** (Top) UMAP projections of per-token susceptibilities for all heads across training for seed 4. Arrows point from the posterior to the anterior. (Bottom) Per-token susceptibilities for seed 4 are aggregated by pattern and their values are averaged and plotted over the course of training, using approx. 2M tokens. On the left half (gray outline) are the per-head, per-pattern susceptibilities for layer 0, while layer 1 susceptibilities are on the right (black outline).

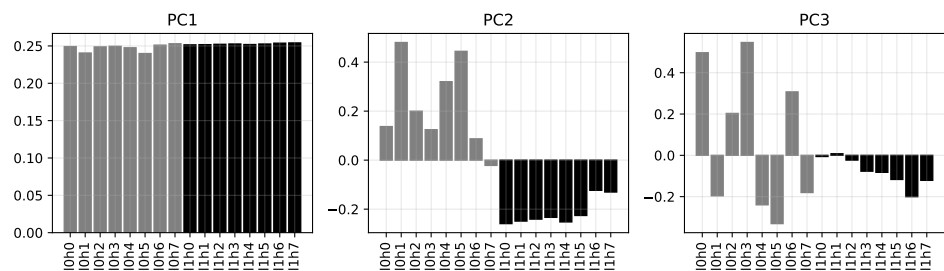

Figure 15: **Head loadings of per-token susceptibility PCA for seed** 1. In Wang et al. (2024, Appendix G) it was found that in this seed the previous-token heads are 0:1 and 0:4, the current-token head is 0:5 and the induction heads are 1:6 and 1:7.

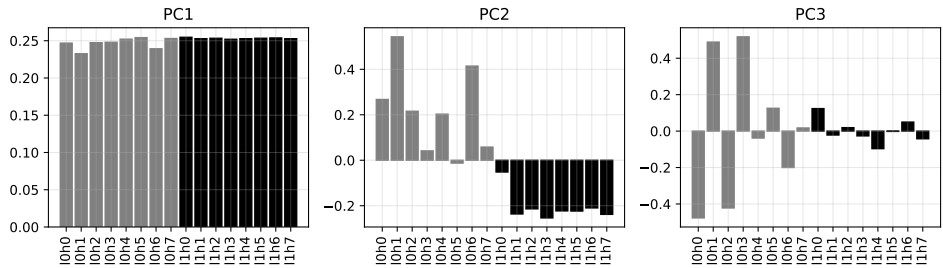

Figure 16: **Head loadings of per-token susceptibility PCA for seed** 2. In Wang et al. (2024, Appendix G) it was found that in this seed the previous-token head is 0:1, the current-token head is 0:6 and the induction head is 1:0.

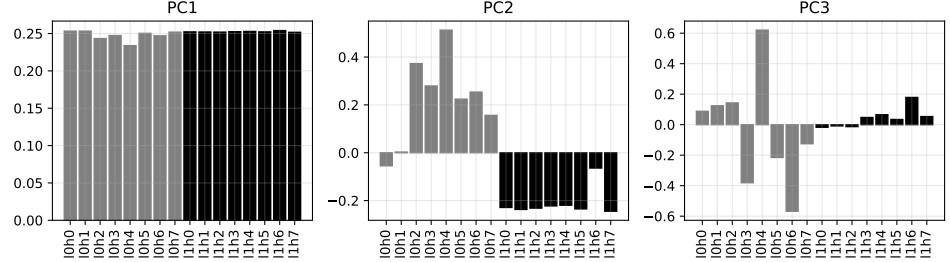

Figure 17: **Head loadings of per-token susceptibility PCA for seed** 3. In Wang et al. (2024, Appendix G) it was found that in this seed the previous-token head is 0:4, the current-token head is 0:2 and the induction head is 1:6.

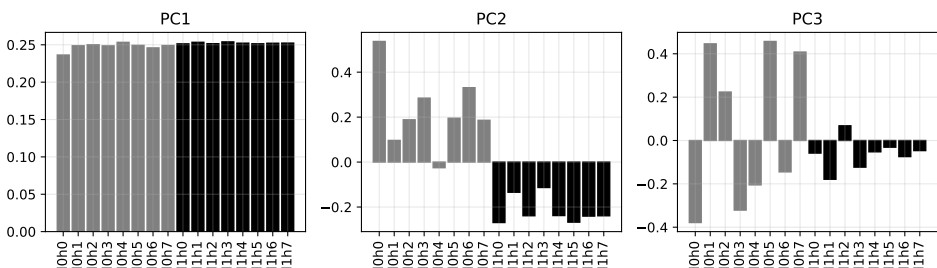

Figure 18: **Head loadings of per-token susceptibility PCA for seed** 4. In Wang et al. (2024, Appendix G) it was found that in this seed the previous-token heads are 0:0 and 0:3, the current-token head is 0:6 and the induction heads are 1:1 and 1:3.

Table 2: Details of the Pile (Gao et al., 2020) subset datasets used in our analysis.

| Dataset | Description | Size (Rows) |
|---|---|---|
| GITHUB-CODE | Code and documentation from GitHub | 100k |
| PILE-PILE-CC | Web crawl data from Common Crawl | 100k |
| PILE-PUBMED_ABSTRACTS | Scientific abstracts from PubMed | 100k |
| PILE-USPTO_BACKGROUNDS | Patent background sections | 100k |
| PILE-PUBMED_CENTRAL | Full-text scientific articles | 100k |
| PILE-STACKEXCHANGE | Questions and answers from tech forums | 100k |
| PILE-WIKIPEDIA_EN | English Wikipedia articles | 100k |
| PILE-FREELAW | Legal opinions and case law | 100k |
| PILE-ARXIV | Scientific papers from arXiv | 100k |
| PILE-DM_MATHEMATICS | Mathematics problems and solutions | 100k |
| PILE-ENRON_EMAILS | Corporate emails from Enron | 100k |
| PILE-HACKERNEWS | Tech discussions from Hacker News | 100k |
| PILE-NIH_EXPORTER | NIH grant applications | 100k |
| PILE1M | Combined samples from all subsets | 1M |

# H  ADDITIONAL EXAMPLES OF TOKENS

In Figure 19, we present the seed 1 UMAP using two additional projections.

Comparing with Figure 5 we can see how the linear structure in the data matrix exhibited by PCA is reflected also by the (non-linear) projection computed by UMAP, and how some of the features in the UMAP are intrinsically non-linear (e.g. variations that occur away from the origin). Note that the separation between word ends (blue) and induction patterns (orange) which characterizes PC2 Figure 5 is mostly a phenomena that we see in *positive PC1* in the UMAP. Also, the streak of numeric tokens (yellow) visible in the UMAP (particularly in the XY view) appears only in PC5.

In Figure 19 we highlight several *streaks* of tokens, by which we mean sets of tokens following the same token pattern (i.e. having the same color in the UMAP) and which are arranged contiguously along the anterior-posterior direction. Streak (A) consists of token sequences $xy$ where $x$ ends with the conclusion of a sentence and $y$ is a word start token which decodes to a space followed by a capitalized word, as shown in Figure 20. Other token sequences involve $y$ among `The`, `This`, `While`, `These`, `There`. Streak (B) consists of numeric tokens, largely appearing in math questions from DM_MATHEMATICS. Streak (C) consists of spacing token sequences $xy$ where $y$ is a tab `\t` (which appear in code and HTML). Streak (D) consists of `the` tokens. Streak (E) consists of `to` and `and` tokens.

Observe that streaks (D), (E) consist of *function words* `the`, `to`, `and` which appear on the dorsal spine of the UMAP, opposite to induction tokens which are more often "content" words. It is interesting that as we move along the dorsal spine of the UMAP from the posterior, we encounter space tokens ` ` then newlines `\n` (see Figure 8) and then eventually these function words; from some perspective these are all "structural" tokens.

# I  ADDITIONAL RELATED WORK

**Development of the induction circuit.**     The development of the induction circuit in the same language model used in this paper was studied in Hoogland et al. (2025, §4). The emergence of the induction circuit was tracked with prefix scores for the two induction heads and the ICL score of the network, while in Wang et al. (2024, Fig. 7) the emergence was studied using K-composition between the induction heads and previous-token heads. According to the prefix and ICL scores, most of the change occurs over LM4, whereas the K-composition increases more gradually from around 2000 steps, although with a step change in slope around the LM3-LM4 boundary (8500 steps). In general, LM3 and LM4 were identified as the stages where the induction circuit develops.

**Structural inference for the small language model.**     In Baker et al. (2025) it was found that if PCA is computed for each layer separately (in the sense that we take two data matrices, one with a column per head in layer 0 and the other with a column per head in layer 1) then PC2 for both matrices has a strong positive loading on the heads $\{$ 0:1 , 0:4 , 0:5 $\}$ and $\{$ 1:6 , 1:7 $\}$ respectively. That is, the heads we would expect based on the idealized description to be involved in the induction circuit, together with the current-token attention head 0:5 . Moreover, this is true for all four seeds. In this sense, structural inference as introduced in Baker et al. (2025) finds the induction circuit, which we define to include the current-token attention head.

**Thom's morphogenesis and structural stability.**     It was the proposal of Thom (1972) that the stages in embryonic development are associated with catastrophes in the sense of singularity theory. The susceptibilities $\chi$, which are defined as integrals against the quenched posterior $\exp(-nL(w))\varphi(w)$, are sensitive to the geometry of the level sets of $L(w)$ (Arnold et al., 1985). We conjecture that the qualitative changes in the pattern of susceptibilities, which take visual form in the evolution of the UMAPs of this paper, are associated with specific changes in this geometry. Under this hypothesis the development of shape and form that we study in the UMAP in this paper is close to the vision of morphogenesis put forward by Thom.

**Data manifold.**     A fundamental insight in deep learning is that learning representations is closely related to learning a coordinate system on the "data manifold," see Bengio et al. (2013, §8) and Fefferman et al. (2016). If we think of the UMAP as a representation, from the model's point of view, of the data samples as a manifold (with boundary), then it is natural to think of the geometry of the configuration of tokens (especially at the boundary) as being meaningfully connected to the learned

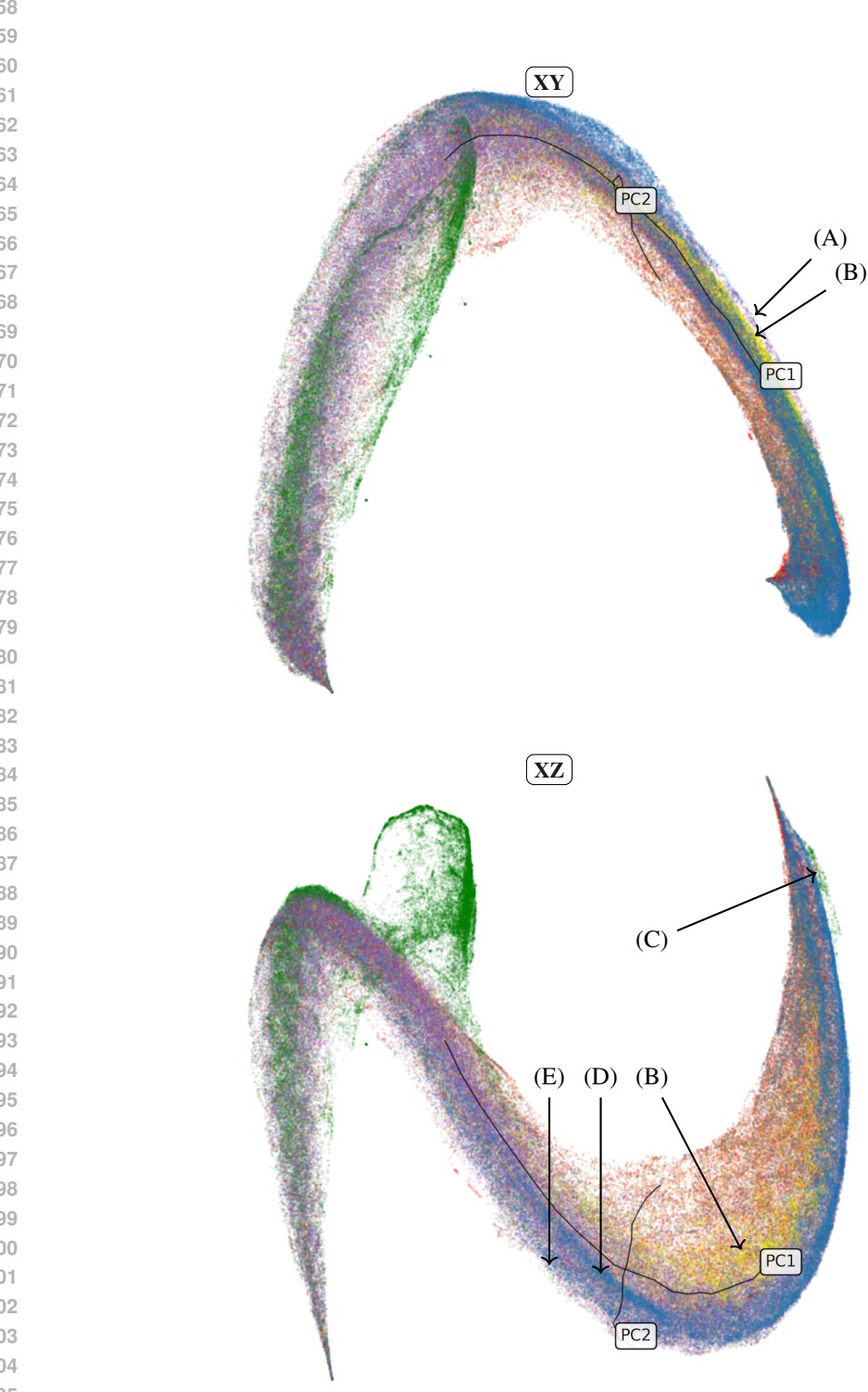

Figure 19: **The rainbow serpent, other angles.** UMAP projection of per-token susceptibilities at the end of training, showing two additional 2D projections (the YZ projection was shown in the main text). Shown are lines corresponding to the first two principal components (labels PC1, PC2 occur at the positive end of the axis). We highlight token streaks (A)-(E) which are detailed in Figure 20.

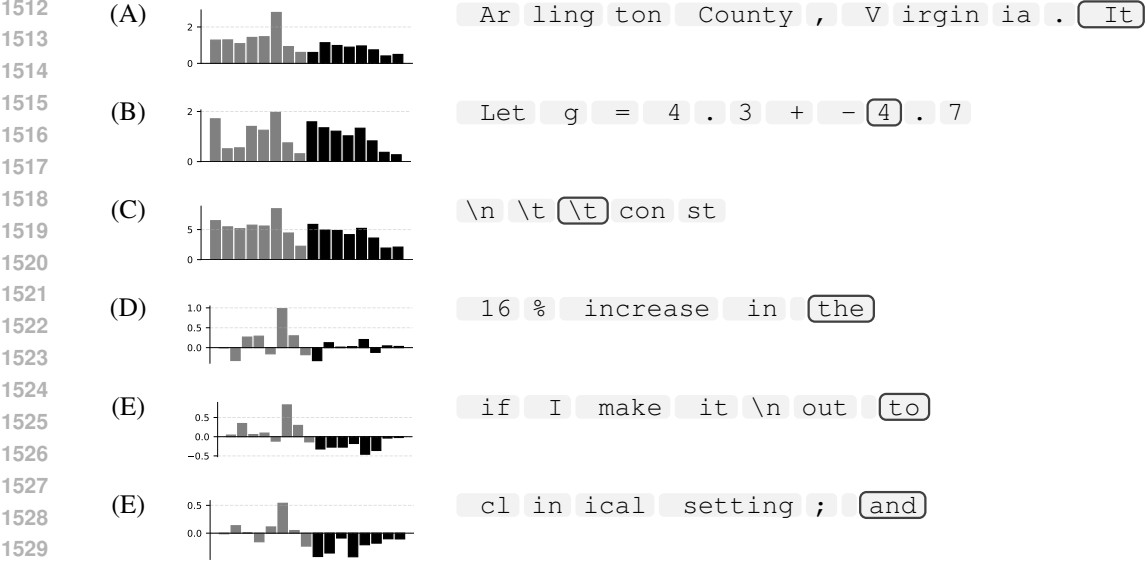

Figure 20: **Token streaks.** Susceptibility vector $\eta_w(xy)$ showing each head in order 0:0-0:7 (gray) 1:0-1:7 (black) and the token $y$ (black outline) in its context $x$. The examples are taken from the streaks (A)-(E) identified in Figure 19. Note the similarity of the susceptibility vectors of the tokens the , to , and .

representation within the model. It would be interesting to study more deeply the relation between *representation geometry* and *loss landscape geometry* that appears here.

**Structural inference vs circuit discovery.** The approach to identifying computational structure in neural networks taken in Baker et al. (2025) and continued here views structure as being about *coherent patterns of variation in the responses of model components to particular patterns in the input*. As a methodology, this is almost identical to the way structure is defined in modern genomics. Biologists identify functional "modules" or "pathways" by finding gene coexpression networks. They apply a stimulus (for example, a drug) to cells and measure the expression levels of thousands of genes. Genes whose expression levels show a coherent pattern of variation (i.e., they are consistently up- or down-regulated together) are grouped into a functional module. In some respects, this is different from the prevailing "circuit-centric" paradigm in mechanistic interpretability. In fact, among the most prominent structures in our sense is the anterior-posterior axis, which is determined by a coherent pattern of suppression or expression across *all* heads, and the spacing fin, which is not obviously encoded by a small subset of heads. We expect that structural inference and circuit-based interpretability are complementary methods: for instance, having identified the spacing fin as an interesting structure by our methods, one could then target it for more mechanistic analysis.

## J STATEMENT ON USE OF LLMs

LLMs were used in the course of this research to support literature review for related work as well as to generate some of the code used to implement experiments and to analyze and plot the data. All LLM-generated output was reviewed by a human author.

