# OpenReview forum: "Embryology of a Language Model"
_ICLR.cc/2026/Conference — ICLR 2026 Conference Withdrawn Submission_

### Official Review · Reviewer_6eSy · 2025-11-01

**Soundness:** 3
**Presentation:** 3
**Contribution:** 3
**Rating:** 6
**Confidence:** 3

**Summary:**

The paper proposes an "embryological" lens on the training dynamics of a 3M-parameter, 2-layer attention-only language model.
The key idea is to compute **per-token susceptibilities** for each attention head, stack them into vectors (\eta_w(xy)), and then visualize these vectors with UMAP over the course of training.
The authors report (i) a long thin "rainbow serpent" manifold whose first two principal axes align with global expression/suppression (PC1) and a dorsal–ventral stratification tied to the induction circuit (PC2), and (ii) a newly described **"spacing fin"** associated with sequences of spacing tokens and their counts.

**Strengths:**

**Originality.**

* Using **susceptibility vectors** (rather than activations) to visualize response is an interesting perspective that complements circuit-centric methods. The "spacing fin" is a surprising, concrete emergent structure that the authors unraveled with their visualization.
* The biological metaphor (anterior–posterior / dorsal–ventral axes) is consistant through the manuscript and helps organize observations about stratification by token pattern.

**Quality / Technical soundness.**

* The susceptibility definition and its sign interpretation (expression vs. suppression) are clearly stated, with an explicit covariance-based definition (Def. 2.1) and discussion.
* Cross-seed visualizations (Appendix F) support claims.

**Clarity.**
The paper is easy to follow, generally well written, figures are plentiful and annotated.

**Weaknesses:**

**The probabilistic setup and tractability of the quenched posterior need more transparency.**

* Eq. (2) introduces the posterior ( $ p^{\beta}\_{n}(w) \propto \exp \\{-n \beta L(w) \\} \phi(w) $ ) with normalizer ( $ Z^{\beta}\_{n} $ ). The *practical* tractability of ( $ Z^{\beta}\_{n} $ ) and how its intractability propagates (or cancels) in ( $ \chi $ ) estimates are not discussed.

**Motivation for Def. 2.1 could be surfaced earlier.**
The definition of susceptibility (Def. 2.1) appears before an intuitive build-up of why *this* covariance captures "expression/suppression." Consider moving the intuitive paragraphs ("Negative susceptibility… Positive susceptibility…") directly before the formal definition illustrating sign and magnitude.

**Head labeling vs. permutation equivariance.**
Section 3 states (based on prior work) which heads are previous-token/current-token/induction (e.g., 0:1, 0:4, 0:5, 1:6, 1:7), but attention heads are *a priori* permutation-equivariant under reindexing. Please add a sentence clarifying **how** heads are *identified and matched across runs/checkpoints*, and how this resolves the labeling issue across seeds.

**Figures / Typo.**

* Fig. 2’s subplots don’t share a y-axis within pattern groups, making comparisons harder. Please share y-axes across rows where meaningful or include small multiples with identical scales. Also label heads (l:h) more prominently.
* Minor: page 9 line ~450 "exhibition / exhibition" → "excitation / inhibition." (If "exhibition/inhibition" is deliberate, please justify the terminology.)

**"Universal body plan" is stated strongly given one architecture/scale.**
The paper emphasizes universality across seeds of *one* tiny attention-only model with a specific tokenizer. Please temper claims or add more evidence: (i) a second tokenizer, or (ii) a small MLP-augmented transformer at the same scale, or (iii) at minimum, a dataset ablation showing how pattern frequencies (Fig. 3) modulate the observed geometry. Even a compact sensitivity table would help.

**Questions:**

**Suggestions:**

* Section 4.1 uses "Serpent" metaphors; we believe the term "Eel" (which has fins) could be better suited.
* On p. 6 you refer to "PC1/PC2" without first saying you did PCA; add a forward pointer to Appendix C.
* On p. 9, line 450, typo "exhibition / exhibition" $\rightarrow$ "exhibition / inhibition".

In addition to the concerns raised in the weakness part.

---

> ### Author Response · Authors · 2025-11-25
>
> Thank you for your review, and for your suggestions on improvements to the paper.
>
> We will opt not to respond line by line to each of the suggestions, but we happily take them all under consideration for a future draft.

---

### Official Review · Reviewer_eDxW · 2025-11-01

**Soundness:** 2
**Presentation:** 2
**Contribution:** 2
**Rating:** 2
**Confidence:** 3

**Summary:**

The paper presents an “embryology” lens for LM training by projecting per-token susceptibility vectors into 2D with UMAP, then tracking how the geometry evolves over training. This is meant to shed light on how language models develop their internal computational structure. The authors find the projection produces a striking rainbow serpent structure whose axes point towards the emergence of an induction circuit.

**Strengths:**

- The approach is unlike most views i've seen in terms of interpreting LLMs and is creative/novel.
- The paper presents a fairly holistic view of interpretability in LLMs. Instead of focusing on single circuits, the method reveals global organization and complementary expression/suppression roles across heads.
- Joint use of UMAP snapshots and per-pattern susceptibility trajectories might point to plausible temporal causal structure emergence.

**Weaknesses:**

- The results presented in the paper are entirely based on  a 3M, 2-layer attention-only model. It’s unclear whether the serpent structure and spacing fin persist or change in mid/large LMs with MLPs and modern tokenizers (e.g., non-whitespace-heavy merges).
- Although partly addressed, the method still relies on a nonlinear, stochastic embedding with known global-distance distortions; the work would benefit from corroboration via isometry-aware metrics in the original space.
- The approach may be sensitive to confounding, since the prominence of spacing tokens may be an artifact of the truncated GPT-2 vocab and dataset composition; more direct controls or alternate tokenizers would help.
- Overall, apart from the PC2 thickening statistic for induction, much of the case rests on visuals. More numerical results in the form of for instance statistical tests (e.g., separability indices, cluster stability, supervised recovery of pattern labels from η) would strengthen claims.

Suggestions for improvements:
- Replicate results on other architectures (with ablations) on a small-MLP transformer and a ~100–300M LM with modern BPE or unigram LM tokenizers;
- Provide chain-to-chain variance, R-hat-style diagnostics, and sensitivity to SGLD hyperparameters; test subsampling stability of η-space geometry.

**Questions:**

1. What exactly does susceptibility represent causally? Is χ interpretable as a local sensitivity to intervention on a head, or more akin to covariance with a loss gradient?

2. Why is the “embryology” metaphor meaningful beyond aesthetics? What else does it add in this context that we would not be able to infer otherwise?

3. Do we expect a similar 2D structure if we used t-SNE, Isomap, or diffusion maps? Is there theoretical reason (e.g., low intrinsic dimensionality of susceptibility space) to expect such a compact geometry?

4. How stable are susceptibility estimates? Did you try random seeds, sampling noise, SGLD parameters (β, ε, γ)? Is there a confidence measure per χ that could be visualized (e.g., variance across samples)?

5. Could susceptibility vectors serve as features for causal discovery (e.g., learning directed edges between heads)?

---

> ### Author Response · Authors · 2025-11-25
>
> Thank you for your review, and for your comments on the novelty of the work and suggestions for improvement.
>
> Regarding the impact of the choice of model and tokenizer:
>
> * As you note, the specific influence of this choice that we speculate on is related to the way in which spacing tokens are tokenized with a truncated tokenizer.
> * We agree that this likely has a strong influence on the presence of a spacing fin in this particular model, but we do not see this as “confounding”. Our precise claim about the methodology we introduce is that for a given training setup, this methodology is capable of reliably picking up interesting structures related to that *particular* training setup. We see it as a strength that, in a case where the model has a particular reason to care about consecutive spacing tokens and has some structure relating to that, that the methodology picks that up, but do not claim that such structure exists in *general*, across other choices of training setup.
> * Our preliminary results on larger models (up to O(1B) scale) with a full-size tokenizer and MLP layers still appear to have similar clusters or structures visible in the UMAP visualization, although the cluster of spacing tokens in those cases are not clearly about counting consecutive spaces, but rather may have different mechanistic explanations (in these cases, there appears to be a significant amount of shared structure across model scale for a fixed tokenizer).
>
> Regarding additional statistical support for the claims:
>
> * We do have some such data available (sampling chain variance, SGLD hyperparameter sensitivity, subsampling stability, etc) which we checked carefully in the process of conducting the work but which was overlooked in the appendices. We are happy to incorporate some of this into a new appendix.
>
> Regarding the list of questions:
>
> * A susceptibility is a measurement of how the posterior distribution over weights would change with respect to an infinitesimal change in the data distribution. One could also think of this mechanistically as being a form of ablation on a model component.
> * Thinking within a metaphor like “embryology” can suggest directions and techniques to focus on. Though such a metaphor may not be perfect, as far as such metaphors can be followed, there are many well-trodden paths that can be retraced, and a meaningful analogy provides predictive power for what is worth exploring or not. You may argue that this doesn’t provide much value or that it provides a lot.
> * We considered t-SNE and PCA as alternate dimensionality reduction techniques at various points in the research process. We opted for UMAP due to a variety of tradeoffs, including computational cost.
> * We do not have an explicit, quantitative confidence measure, but we did try all of these suggestions in addition to others not mentioned. This was excluded in the appendix because a proper treatment would involve a significant amount of additional context or setup.
> * This certainly seems plausible, though the most obvious way in which this could be done would involve checking correlations between heads on specific tasks or data samples, and thereby dramatically reducing the search space.

---

### Official Review · Reviewer_Esoa · 2025-11-05

**Soundness:** 3
**Presentation:** 3
**Contribution:** 2
**Rating:** 4
**Confidence:** 2

**Summary:**

The paper proposes an “embryological” lens on visualization of training small language models: compute a per‑token susceptibility vector (one coordinate per attention head) that measures how changing weights in a component covaries with the loss on a specific continuation, then apply dimension reduction techniques to many such vectors over training. The resulting 2D plots form a characteristic “rainbow serpent” whose long axis (PC1) reflects global expression vs. suppression across heads and whose short axis (PC2) thickens as an induction circuit emerges. The authors also report a previously unremarked structure, a “spacing fin”, consisting of spacing tokens, especially those preceded by long runs of spaces/newlines, which they hypothesize reflects counting of spacing tokens.

**Strengths:**

Overall, I believe the authors have studied an interesting question and created various intriguing visualizations that could be of interest to the community. The paper presentation was good, and the figures are very nicely presented. In particular,

1. The "embryological" analogy, framing model training as a developmental process, is conceptually intuitive yet powerful. Applying UMAP to susceptibility vectors, rather than just model activations, is a novel approach. It provides a global visualization of how the entire set of model components (attention heads) collectively organizes to handle different token patterns.

2. The theory was validated (at least in small scales) nicely by the experiments showing the emergence of induction circuits, which were intensively studied in literature, and placing it into their own contexts, offering their interpretation of the underlying mechanisms of the model.

**Weaknesses:**

The major limitations are already straightforwardly discussed in the paper. Here I rephrase the two most important ones in my opinion:

1. The experiments are pretty severely limited by scale, as they were only visualized with a tiny 3M model with two layers of attention-only modules. It is a pretty significant leap from even tiny-scaled language models by today's standards. This is (in my opinion) the most significant weakness of the present paper, and it is not clear what the limitation is for the authors not to report more extensive experiments.

2. Some key claims (e.g. the spacing fin’s separation) depend critically on UMAP. The paper also notes PCA fails to reveal the fin in the first three PCs and suggests it lives in higher PCs. However, as the authors have pointed out, UMAP is a non-linear visualization technique that can distort global geometric structures. This raises the question of the robustness as well as reliability of their findings when migrating to different architectures, as well as making interpreting the results harder.

A minor point is that a lot of biology references are made, which may make it hard for someone without knowledge of biological sciences to capture the analogies. Despite the above, I still have questions for the paper (see below) and will consider raising the score if the authors provide satisfying responses to the weaknesses/questions.

**Questions:**

1.  Following up with the scalability concern, how much computational overhead is there to compute the necessary statistics for the proposed visualization? Is training a larger base model actually the biggest computational burden, or are the tools developed for visualization demanding nontrivial computational resources beyond training the model?

2. The authors discussed the experiment results, "likely influenced by the tokenizer, and different tokenization strategies could lead to different learned structures." Can you provide more empirical evidence or insights into why this might be the case, or better, provide experiments on how tokenization changes the results?

3. While the figures are interesting, I didn't find much discussion on the impact for practitioners who are training a model or understanding a trained model. What qualitative or quantitative features of the proposed framework (e.g., emergence of induction circuits) are expected to transfer when training a (possibly much different) model or inspecting a trained model?

---

> ### Author Response · Authors · 2025-11-25
>
> Thank you for your review and for your questions.
>
> Regarding scalability concerns and computational overhead:
>
> * The computational overhead to computing the necessary statistics scales somewhere between logarithmically and linearly with model size. We estimate that at the O(1B) parameter scale, the computational cost is somewhere around 10% of the training cost. We expect that this cost can be significantly reduced as well, the methodology in this paper takes a relatively naive / brute force approach to computing all of the statistics, and there is plausibly at least an order of magnitude of low hanging fruit for improving the efficiency.
> * Regarding reporting results on a smaller model rather than a larger one, or lacking more extensive results: as model size increases, one can be less certain of what is really happening internally. Even in a small language model of this size, which has been studied extensively in earlier papers, an improved microscope reveals new details and understanding. We therefore chose to focus closely on a smaller model to demonstrate that the methodology works when grounded by a setting that is relatively well understood before moving to larger models. The transition to larger models will be the focus of future work, which we have repeated the methodology on and are currently studying.
>
> Regarding the impact of the tokenizer:
>
> * Models learn patterns in the data, but the patterns in the data the model sees are first filtered through the tokenizer, and so in this sense it is natural that the tokenization should affect the structure learned by the model to predict the given patterns. The specific influence that we speculate on here is related to the vocab size of the tokenizer. This particular model uses the GPT-2 tokenizer truncated to 5000 vocab words. In the original tokenizer, later vocab words include many more common words paired with spaces, or longer sequences of consecutive spaces as a single token. As a result, individual space tokens are less common in the tokenized data in the original tokenizer, while they are more common in the small model we study in this work. Therefore, we expect that the specific structure investigated in the spacing fin for this model is influenced by the fact that this model simply sees many more space tokens than a language model trained with a typical 50k+ length vocabulary.
> * Our preliminary results on larger models with a full-size tokenizer still appear to have similar clusters or structures visible in the UMAP visualization, although the cluster of spacing tokens in those cases are not clearly about counting consecutive spaces (we do not claim that all models should have such a spacing fin indicating structure for counting consecutive spaces, only that our methodology is capable of reliably picking up whatever such structures models do learn, for their given task and training setup).
>
> Regarding practical applications:
>
> * In terms of understanding a given trained model, at a coarse level, our methodology (and further developments of it) offers a way to automatically discover interesting structure in a model or to understand how a model “sees” the data. While the structures focused on here (the induction circuit or spacing fin) may be relatively simple, this is a feature of the model in question being relatively simple (which we claim is desirable as a first step in developing a new methodology).
> * In the future, we may see that this allows us to study a larger, more complicated model, and to observe the organization of more abstract or practically relevant behaviors. For example, if chat models in production have some undesirable behavior, we may be able to automate the discovery of inputs that elicit this behavior. This type of understanding may then naturally lead to training interventions.

---

### Official Review · Reviewer_7Ksj · 2025-11-12

**Soundness:** 2
**Presentation:** 3
**Contribution:** 1
**Rating:** 0
**Confidence:** 4

**Summary:**

This paper introduces an "embryological" approach for studying the development of structure during the training of a small language model. The authors use UMAP on per-token susceptibility vectors to visualize how structural organization emerges through training.

**Strengths:**

**Originality: poor**
Application of UMAP to high-dimensional susceptibility vectors for visualization is a straightforward combination of existing methodologies rather than a substantial advance. The framing of “embryology” and “body plan” in neural networks reads more as metaphorical novelty than genuine technical originality.

**Quality: poor**
The overall quality of the work is limited by a lack of rigorous experimental validation and an overreliance on qualitative visualization. Key claims about the “universality” and interpretability of observed structures are not robustly substantiated, and methodological choices (model size, tokenizer, architecture) are unjustified and insufficiently explored. Findings appear sensitive to experimental setup and dimensionality reduction parameters.

**Clarity: good**
The paper is generally clear and well-organized, with visualizations that are appealing and easy to follow. Explanations of the experimental setup and the visualization process are accessible, and the writing is coherent.

**Significance: poor**
The significance is limited, as the claims do not meaningfully advance our theoretical or practical understanding of neural network internals. The scientific insights derived from the visualizations are superficial. The potential for broader impact is unclear given the narrow experimental scope.

**Weaknesses:**

**Unsound/superficial application of UMAP.**
- The paper relies heavily on UMAP for what amount to no more than qualitative visualizations. The limitations of UMAP for interpretability are identified in the appendix, and others are well-documented in various literatures. As a consequence, "anatomical" claims lack quantitative rigor to support the "serpent" as a stable, robust feature rather than a visualization artifact. The authors acknowledge that the geometry of the "serpent" should be "nterpreted cautiously." Given that the paper's claims rely almost exclusively on this geometry, all results should be interpreted *at least* as cautiously.
- The paper's claim that UMAP "faithfully represented" aspects of the underlying high-dimensional distribution is not justified. There is little to no effort to quantify the claims of reliability or interpretability of the UMAP projections.
- The authors remark that UMAP parameters were varied and patterns that did not persist were dismissed. However, there is no quantitative stability result to support that the observed structures are not induced by specific hyper-parameter choices.

**Experimental limitations and poor generalization claims.**
- Critical aspects like the "spacing fin" are demonstrated to be contingent on the tokenizer and possibly the dataset (?). But the paper does not explore how changing tokenization, data distribution, or model size / complexity alter these findings. Again, the paper's claims to have identified a persistent structure but makes no effort to rigorously test or quantify this persistence across variations. Claims to "universality" of the body plan are called into question.
- Findings are only supported by UMAP *visualizations* and not quantitative measurements.
- The "spacing fin" is presented as a newly discovered structure, but its mechanistic significance is left at best conjectural.

**Questions:**

- Quantitative support for persistence or robustness of the "body plan" and "spacing fin"?
- Any sort of rigorous quantitative evidence that the observed structure are not artifacts of UMAP?
- Given the dependence on tokenizer, do similar features emerge with alternative tokenization schemes and different data distributions?

---

> ### Author Response · Authors · 2025-11-25
>
> Thank you for your review, and for surfacing opportunities for us to improve the communication of our work.
>
> Regarding the work being “a straightforward combination of existing methodologies”. Susceptibilities for neural networks are far from a well-established methodology (this being only the second paper in which they appear), and so we believe further development and evidence of their utility constitutes a novel contribution.
>
> Regarding the reliance on qualitative visual features of UMAP: use of UMAP for dimensionality reduction and visualization is a commonly used, valid methodology applied to large fractions of modern quantitative science. This depends on having applied an appropriate standard of caution, which is where we read most of the disagreement as coming from. We perhaps hedged too much in the direction of caution in our writing, but we push back strongly against the claim that our “findings are only supported by UMAP *visualizations* and not quantitative measurements.” We highlight examples of particular quantitative measurements that support claims mentioned in the review:
>
> * Regarding the body plan: while something like the curl of the serpent may indeed be a visualization artifact, we are careful to provide quantitative evidence that the original, high dimensional data is in fact long and thin. We do so by pointing out that PC1 explains over 95% of the variance, as well as quantitatively demonstrating the dorsal-ventral stratification by examining PC2 in Figure 7, and both PC1 and PC2 can be seen overlaid on the body of the serpent in Figure 1\. We repeat most of our analysis on additional training seeds in the appendix. Although we did not include the PC1/PC2 values there, these were very stable results.
> * Regarding the spacing fin and other structures discussed in the paper, we provide more concrete quantitative data in Figure 2, Figure 5, Figure 6, Figure 8, and Figure 11\.
>
> While one may argue whether these are *sufficient* as supporting evidence (we of course argue that they are), it is false to claim that there is *no* supporting evidence of this nature.
>
> Regarding the stability of UMAP across hyperparameters:
>
> * We acknowledge that we do not include a quantitative stability result here. However, we swept over what we understand to be a very wide range of hyperparameters for the dataset that we were working with, more or less across the entire set of reasonable hyperparameters, and so the claim that we are only considering structures that are stable under those conditions is stronger than it may first appear. We may include example results from those hyperparameter sweeps in a future draft, but we are confident that upon seeing them that it would seem obvious that these are not induced by specific hyperparameter choices.
>
> Regarding the dependence on tokenizer and a related comment on “universality”:
>
> * All language models and their corresponding internal structures are dependent on the tokenizer; the way patterns in the data are processed by the model pass through the tokenizer and the model is trained on this. This is true of all interpretability papers on language models, and yet it is not standard to compare results across different tokenizers in those cases.
> * In our case, we speculate on the impact of the tokenizer because it is more different than usual (in terms of having smaller vocab size). We do not assert that the spacing fin is important in and of itself, but rather we see the spacing fin as evidence that our methodology reliably finds structure induced by the particular training set up used in this work (we do this by checking other seeds with the same setup – it’s not *a priori* obvious that this should work). This is then evidence that our methodology would reliably find (not necessarily the same) structure in other models with different training setups. This is the sense in which we claim our methodology is universal, not that the specific structure we find is.

---

### Note · Authors · 2025-11-25

I have read and agree with the venue's withdrawal policy on behalf of myself and my co-authors.